


# Aerosols from anthropogenic and biogenic sources and their interactions: modeling aerosol formation, optical properties and impacts over the central Amazon Basin

Janaína P. Nascimento[1], Megan M. Bela[5,6], Bruno Meller[2], Alessandro L. Banducci[8], Luciana V. Rizzo[7], Angel Liduvino Vara-Vela[4], Henrique M. J. Barbosa[2], Helber Gomes[9,10], Sameh A. A. Rafee[3], Marco A. Franco[2], Samara Carbone[11,2], Glauber G. Cirino[12], Rodrigo A. F. Souza[1], Stuart A. McKeen[5,6], and Paulo Artaxo[2]

[1]Post-graduate Program in Climate and Environment (CLIAMB), National Institute for Amazonian Research and Amazonas State University, Manaus, AM, Brazil
[2]Institute of Physics, University of Sao Paulo, Sao Paulo, SP, Brazil
[3]Department of Atmospheric Sciences, Institute of Astronomy, Geophysics and Atmospheric Sciences, University of São Paulo, SP, Brazil
[4]Center for Weather Forecasting and Climate Studies, National Institute for Space Research, Cachoeira Paulista, Sao Paulo, SP, Brazil
[5]Cooperative Institute for Research in Environmental Sciences, University of Colorado, Boulder, CO, USA
[6]NOAA Earth System Research Laboratory, Boulder, CO, USA
[7]Department of Environmental Sciences, Institute of Environmental, Chemical and Pharmaceutics Sciences, Federal University of Sao Paulo, Sao Paulo, SP, Brazil
[8]Department of Physics, Colorado State University, Fort Collins, CO, USA
[9]Institute of Atmospheric Sciences, Federal University of Alagoas, Maceió, AL, Brazil
[10]Department of Meteorology, Federal University of Campina Grande, Campina Grande, Brazil, PB, Brazil
[11]Federal University of Uberlândia, Uberlândia, MG, Brazil
[12]Department of Meteorology, Geosciences Institute, Federal University of Pará, PA, Brazil

**Correspondence:** J. M. P. Nascimento (janaina@if.usp.br)

**Abstract.**

   The Green Ocean Amazon experiment - GoAmazon2014/5 explored the interactions between natural biogenic forest emissions from Central Amazonia and urban air pollution from Manaus. Previous GoAmazon2014/5 studies showed that nitrogen oxides ($NO_x$ = NO + $NO_2$) and sulfur oxides ($SO_x$) emissions from Manaus strongly interact with biogenic volatile organic

5   compounds (BVOCs), affecting secondary organic aerosol (SOA) formation. In previous studies, ground based and aircraft measurements provided evidence of SOA formation and strong changes in aerosol composition and properties. Aerosol optical properties also evolve, and their impacts on the Amazonian ecosystem can be significant. As particles age, some processes such as SOA production, black carbon (BC) deposition, particle growth, and the BC lensing effect change the aerosol optical properties, affecting the solar radiation flux at the surface. This study analyzes data and models SOA formation using the Weather

10   Research and Forecasting with Chemistry (WRF-Chem) model to assess the spatial variability of aerosol optical properties as the Manaus plumes interact with the natural atmosphere. The following aerosol optical properties are investigated: single scattering albedo (SSA), asymmetry parameter ($g_{aer}$), absorption Ångström exponent (AAE), and scattering Ångström exponent





(SAE). These simulations were validated using ground based measurements at three experimental sites: Amazon Tall Tower Observatory - ATTO (T0a), downtown Manaus (T1), Tiwa Hotel (T2) and Manacapuru (T3), as well as the G1 aircraft flights. WRF-Chem simulations were performed over seven days during March 2014. Results show a mean biogenic SOA (BSOA) mass enrichment of 512% at the T1 site, 450% in regions downwind of Manaus such as the T3 site and 850% in areas north of the T3 site in simulations with anthropogenic emissions. The SOA formation is rather fast, with about 80% of the SOA mass produced in 3-4 hours. Comparing the plume from simulations with and without anthropogenic emissions, SSA shows a downwind reduction of approximately 10%, 11% and 6% at the T1, T2 and T3 sites, respectively. Other regions, such as those further downwind of the T3 site, are also affected. $G_{aer}$ values increased from 0.62 to 0.74 at the T1 site and from 0.67 to 0.72 at the T3 site when anthropogenic emissions are active. During the Manaus plume aging process, a plume tracking analysis shows an increase in SSA from 0.91 close to Manaus to 0.98 160 km downwind of Manaus as a result of SOA production and BC deposition.

## 1 Introduction

Aerosol particles are present in the atmosphere in highly variable types and concentrations, which contribute differently to climate forcing, cloud formation and development, as well as ecosystem impacts. Particles may have a cooling or heating effect on the atmosphere, and their climatic roles are defined by their interactions with solar and terrestrial radiation fluxes, which strongly depend on their optical properties (extinction coefficient, SSA, $g_{aer}$, etc.). Radiation attenuation by atmospheric constituents is described by the radiative transfer equation, which requires information on the intensive and extensive optical properties of particulates and gases (Boucher, 2015). The aerosol's effect on radiation can be direct, semi-direct, or indirect. Direct effects are related to scattering and absorption of solar radiation by aerosol particles. These effects tend to dominate under clear sky conditions. Indirect effects involve the aerosol influence on cloud formation and development through cloud droplet activation via cloud condensation nuclei (CCN) (Haywood and Boucher, 2000).

Recent studies in Amazonia that integrated data from ground-based sensors (e.g., Martin et al., 2016; Rizzo et al., 2013; Andreae et al., 2015; Artaxo et al., 2013) with regional numerical simulations (e.g., Rafee et al., 2017; Shrivastava et al., 2019; Medeiros et al., 2017) advanced our understanding of the interactions in Amazonia background aerosol urban anthropogenic emissions. However, none of these studies have quantified the impact of atmospheric aerosols on the Amazonian radiative forcing.

A previous study conducted over the Amazonian region during the GoAmazon2014/5 experiment found strong SOA production, with an enhancement of BSOA formation in both the Manaus plume and its outflow by a factor of 100–400% on average during the afternoon of March 13, 2014 (Shrivastava et al., 2019). In Southeast Manaus, de Sá et al. (2018) showed an increase in SOA ranging from 25% to 200% under polluted conditions compared to background conditions, including contributions from both primary and secondary particulate matter (PM). All of these studies are related to an idea suggested in Palm et al. (2018), that anthropogenic emissions play a significant role in SOA production. Cirino et al. (2018) indicate that during the dry season an increase of 40% in the mass concentration of organic aerosols is attributed to SOA formation during





transport from Manaus to downwind sites (T2 and T3). Conversely, the same increase was not observed during wet season. The Manaus anthropogenic emissions are rather constant over the year, representing the major influence on the anthropogenic organic aerosol source and contributing to the OA increase downwind of Manaus (de Sá et al., 2018; Shrivastava et al., 2019; Martin et al., 2010).

A possible strategy to improve estimates of the urban plume impact on optical properties downwind of Manaus is to create modeling regional scenarios with and without anthropogenic emissions, and comparing them to analyze how the emissions affect aerosol properties. Other studies have used sensitivity scenarios to understand how aerosol optical properties and secondary formation can be affected by events such as biomass burning (Vara-Vela et al., 2018) or urban pollution (Shrivastava et al., 2019). Many studies have focused on improving the understanding of an urban plume's impact on aerosol optical properties by comparing measurements during background conditions with periods affected by the pollution plume (Palacios et al., 2020; de Sá et al., 2019; Brito et al., 2014; Rizzo et al., 2013, 2011). However, little work has been done to analyze the atmospheric chemistry in the regions typically within the plume but without the plume's effects. This is particularly critical during the wet season, when aerosol levels associated with biomass burning are low and biogenic aerosols become more sensitive to external disturbance. Numerical simulations with high-resolution regional models such as the Weather Research and Forecasting Model with Chemistry (WRF-Chem; Grell et al. (2005)) are necessary for this strategy to quantify the effects of urban areas on aerosol levels and ultimately on the ecosystem, especially in regions that lack ground based observations.

Different aerosol optical properties have been used to study aerosol impacts on ecosystems and the radiation balance, such as SSA (e.g., Dubovik and King, 2000; Lim et al., 2014; Russell et al., 2010; Rizzo et al., 2013), SAE and AAE (e.g., Romano et al., 2019; Palacios et al., 2020), and $g_{aer}$ (Korras-Carraca et al., 2015). The significant impacts of Manaus urban emissions on the characteristics of the aerosol population (size distribution, quantity, chemical and physical composition) in regions downwind of Manaus have been described by Rizzo et al. (2013). However, there are no results considering simulation scenarios when the Manaus pollution plume component is turned on and off.

The objective of this work is to model secondary aerosol formation in Central Amazonia, comparing modeled scenarios with and without anthropogenic emissions, examining the interactions between natural biogenic emissions urban air pollution from Manaus and investigating their impact on aerosol optical properties. We have extensively validated the model predictions with ground-based measurements, and estimate how the optical properties may be affected by the plume aging process (see Fig. 1a). This is the first study to our knowledge that focuses on aerosol optical properties such as SSA, $g_{aer}$, and absorption and scattering coefficients over a geographically extended area over Central Amazonia, using numerical simulations and ground based data.

## 2   Model Description, Emissions and Observations

### 2.1   Study Region and Methodology

The Amazonian region has an annual mean temperature of around 26°C due the intense solar radiation reaching the surface (Nobre et al., 2009), with an annual average precipitation of 2,300 mm year$^{-1}$ (Fisch et al., 1998). In the wet season (between





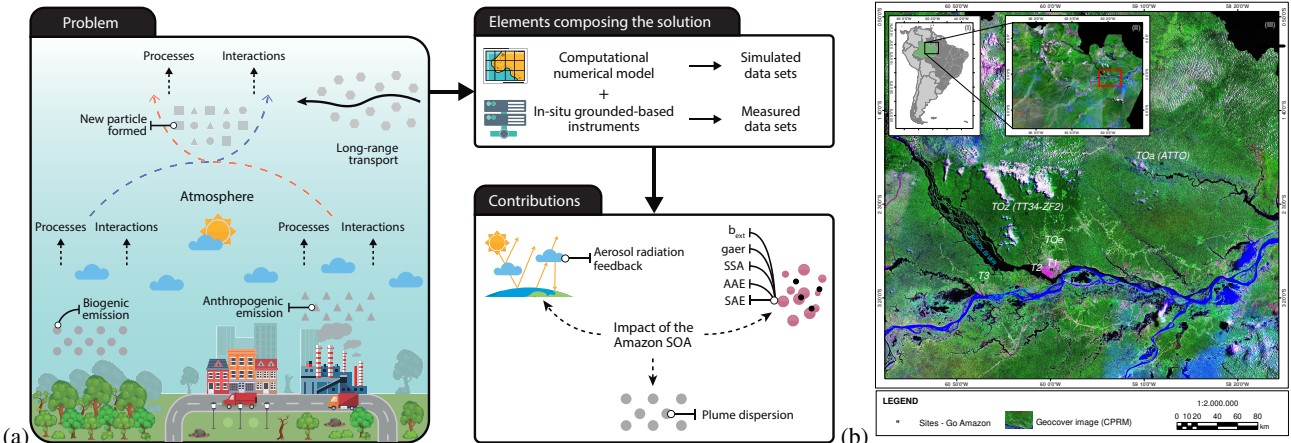

**Figure 1.** (a) Problem: The atmosphere with different pollution and natural sources and the interactions between them. Elements composing the solution: Computational numerical model and observational data sets. Contributions: Understanding the impact of the Manaus air pollution plume on aerosol optical property variability over the Amazon rainforest during the GoAmazon2014/5 experiment. (b) Sampling stations TT34-ZF2 (T0z) (ca. 60 km northwest) and ATTO (T0a) (ca. 150 km northeast), are both upwind of Manaus; Downtown Manaus (T1), Tiwa Hotel (T2) (ca. 8 km direction) and Manacapuru (T3) (ca. 70 km direction), are all downwind of Manaus. (I) South America and Brazil map. (II) Amazon region, the red rectangle indicates the area where the GoAmazon2014/5 experiment meteorology stations were located and the region used for the WRF-Chem simulations.

January and May), with the Inter-tropical Convergence Zone (ITCZ) extending south over Manaus, it is possible to find one

of the lowest particle number concentrations over a continental area in the world (Andreae et al., 2015; Artaxo et al., 1994; Martin et al., 2016). In the wet season, the very high precipitation rate makes it virtually impossible for fires to occur, so the atmosphere is dominated by biogenic emissions, with a episodic component of Sahara desert dust and biomass burning emissions transported from Africa (Artaxo et al., 1990, 1993, 2013; Pöhlker et al., 2018, 2019)

Manaus is a city located in central Amazonia at latitude 3°06'07" and longitude 60°01'30". It currently has a population

of about 1.8 million and an urban area of 377 km$^2$. For this study we focus on a region centered on Manaus extending from latitudes -5.3° S to -0.76° S and longitudes -63.07° W to -56.90° W (see Fig. 1b). This 600 km by 450 km approximately rectangular area comprises the urban area of Manaus, its satellite cities and the surrounding Amazonian forests.

Our WRF-Chem simulation was performed over seven days between 7 and 14 March 2014. This period is part of the wet season in the region (Fisch et al., 1998; Martin et al., 2017). The first day were used as a spin-up period, as such they

are discarded from any analysis. Corrections based on the methodology used in Cosgrove et al. (2003) were applied to the simulated temperature values aiming for better agreement between the topography height represented by the model and the one from the GoAmazon2014/5 experiment site.

The choice of the simulated days was made based upon ground-based data availability, which is necessary to evaluate the performance of the model, and the suggestions of Shilling et al. (2018), Shrivastava et al. (2019) and Martin et al. (2017) which





highlight the March 13, 2014 as a golden day to study the evolution of the Manaus plume as it advected to the surrounding Amazon tropical forest. Our investigation focuses on a detailed analysis of March 13, 2014, because on that day the plume reached regions downwind of Manaus such as the T2 and T3 sites. During this period mostly sunny skies were observed with little or no precipitation and the interference from biomass burning and cloud processing was negligible. We track the simulated Manaus plume as it ages in order to investigate the evolution of optical properties. Different analyses of atmospheric variables

with and without anthropogenic emissions were used to characterize changes in aerosol properties downwind of Manaus due to anthropogenic activity.

To track the plume as it ages, its approximate location and extent over time were determined using the Hybrid Single-Particle Lagrangian Integrated Trajectory (HYSPLIT) model (Draxler, 2007; Stein et al., 2007). Forward trajectories were calculated starting from 8 points at 200 masl in a circle of radius $0.03^o$ ($\sim 3.4$ km) centered on the plume's initial location at 6:00 LT. Winds

and other forcing meteorological fields were taken from our WRF-Chem simulation. Average gas and aerosol concentrations and optical properties were calculated in the volume defined by the maximum and minimum latitude and longitude of the 8 points, and altitude range from 100 to 500m. The averaging region is shown in Figure S12 in the SI, and the altitude of the plume is shown in Figure S14 and S15 in the SI. $\Delta$CO was determined by taking the difference in carbon monoxide (CO) between simulations with anthropogenic emissions turned on and off in the simulated region selected by HYSPLIT. The

tracking plume approach was used to track the plume in other days (10 to 14 March 2014) in order to investigate the change in SOA formation due to different $NO_x$ concentrations. The days other than the $13^{th}$ were not exemplary days for observing the evolution of the Manaus plume due to meteorological factors such as precipitation. Additionally, the plume did not appear until 8 LT. As such, our analysis focuses on March 13, 2014.

## 2.2 WRF-Chem Model Description and Setup

The study region was simulated with the WRF-Chem regional model, version 3.9.1.1 (Grell et al., 2005; Fast et al., 2006) using full coupled and online meteorology, gas-phase chemistry and aerosol feedback. The model grid covered the study region with a horizontal grid spacing of 3 km and nx = 200 and ny = 150 grid points. Vertically, hybrid sigma coordinates were used to split the atmosphere into 51 levels, the bottom 10 within the planetary boundary layer (PBL). Data from the Global Model Data Assimilation System (GDAS), with a horizontal grid spacing of 1° and 26 vertical levels was used for the initial and boundary

conditions of the meteorological variables. Chemistry initial and boundary conditions were provided in 3 hour increments at a horizontal resolution of about 40 km x 40 km with 60 vertical levels from the surface up to 60 km by the European Centre for Medium-Range Weather Forecasts (ECMWF) operational model.

The physics, chemistry and emission options used in this study, as well as their corresponding references, are listed in Table 1. The most significant ones for this application are: the Rapid Radiative Transfer Model for General Circulation Model

applications (RRTMG) scheme for longwave and shortwave radiation (Iacono et al., 2008); the Revised Mesoscale Model version 5 Monin–Obukhov scheme for surface layer (Jiménez et al., 2012); the Unified Noah land-surface model for land surface (Tewari et al., 2004); land use provided by the Moderate-resolution Imaging Spectroradiometer (MODIS) with spatial resolution and 20 different classes; the Yonsei University scheme for the boundary layer (Hong et al., 2006); the Morrison





2-moment scheme for microphysics (Morrison et al., 2009) and the Grell-Freitas ensemble convective scheme (Grell et al.,
2014).

We simulated atmospheric chemistry using the Regional Atmospheric Chemistry Model (RACM) coupled with the Modal
Aerosol Dynamics model for Europe/Volatility Basis Set (MADE/VBS) aerosol scheme, which treats the organic gas/particle
partitioning within a spectrum of volatilities (Ahmadov et al., 2012). The RACM includes 21 stable inorganic species (4
being intermediates), 32 stable organic species (4 of which are primarily of biogenic origin). In addition, RACM includes
237 chemical reactions (23 of which are photolysis). MADE/VBS has an advanced SOA module based on VBS approach
to simulate concentrations of the main organic and inorganic gas/particle partitions within a spectrum of volatilities using
saturation vapor concentrations as surrogates for volatility. It also includes less complex aqueous reactions (sulfate - $SO_4$ and
nitrate - $NO_3$ wet deposition) following CMAQ methodology (Sarwar et al., 2011). MADE/VBS has a four-bin VBS with
the SOA precursor yields based on previous smog chamber studies under both high- and low-$NO_X$ conditions (Murphy and
Pandis, 2009; Ahmadov et al., 2012). Yields are for four volatility bins with saturation concentrations of 1, 10, 100, and 1000
$\mu g\ m^{-3}$, and represent aerosol modes – Aitken (< 0.1 $\mu$ m), accumulation (0.1–1 $\mu$m) and coarse (> 1 $\mu$m).

We used the approach by Fast et al. (2006), according to Mie theory (Mie, 1908), in order to account for aerosol radiative
properties such as absorption and scattering coefficients, SSA and $g_{aer}$. These properties are then transferred to the RRTMG
shortwave radiation scheme in order to calculate the corresponding radiative forcing. In addition, the feedback effects of clouds
on aerosol size and composition via aqueous-phase chemistry (Sarwar et al., 2011) as well as wet scavenging processes (Easter
et al., 2004) are considered.

Simulations were conducted in order to analyze how Manaus emissions affect SOA production and aerosol optical proper-
ties over the Amazon. We considered two scenarios: (i) Manaus on, which represents anthropogenic emissions and background
emissions from initial and boundary conditions; (ii) Manaus off, which represents a background scenario, dominated by bio-
genic emissions, with anthropogenic contribution coming from the boundary conditions.

### 2.2.1 Anthropogenic Emissions

Anthropogenic emissions were calculated using the Rafee et al. (2017) inventory, which considers emissions of all classes of
mobile (light-duty, heavy-duty vehicles and motorcycles) and stationary (thermal power plants (TPPs) and Refineries) sources.
Both components were calculated according to emission factors estimates based on experiments conducted inside road traffic
tunnels in São Paulo (Martins et al., 2006; Sánchez-Ccoyllo et al., 2009; Brito et al., 2013), providing the only vehicle emission
factor measurements available in Brazil. Fine particle matter emission fractionation into size and chemical classes were based
on studies developed for São Paulo (Ynoue and Andrade, 2004; Miranda and Andrade, 2005; Albuquerque et al., 2012).

### 2.2.2 Biogenic Emissions

Biogenic emissions were calculated online using the Model of Emissions of Gases and Aerosols from Nature (MEGAN)
version 2 (Guenther et al., 2006). Based on driving variables such as ambient temperature, solar radiation, leaf area index, and



**Table 1.** WRF-Chem simulations configuration used in this study

| | Simulation time: 2014-3-8 00 UTC to 2014-3-15 00 UTC |
|---|---|
| Attributes | Model configurations |
| Grid resolution | dx = dy = 3 km |
| nx, ny, nz | 200 x 150 x 51 |
| Time step | 10 s |
| Vertical resolution | 51 layers from surface to 100 hPa ($\sim$16 km) |
| Physical options | |
| Radiation | Long/shortwave RRTMG scheme (Iacono et al., 2008) |
| Land surface | Unified Noah land-surface model (Tewari et al., 2004) |
| Surface layer | Revised Mesoscale Model version 5 Monin–Obukhov scheme (Jiménez et al., 2012) |
| Boundary layer | Yonsei University scheme (Hong et al., 2006) |
| Cloud microphysics | Morrison 2-moment (Morrison et al., 2009) |
| Cumulus clouds | Grell–Freitas ensemble scheme (Grell et al., 2014) |
| Chemical options | |
| Gas-phase chemistry | Updated RACM version with chemical reactions for sesquiterpenes (Papiez et al., 2009) |
| Aerosol module | MADE/VBS (Ahmadov et al., 2012) |
| Aerosol activation | Abdul-Razzak and Ghan scheme (Abdul-Razzak and Ghan, 2000) |
| Photolysis | TUV (Madronich, 1987) |
| Meteorological IC and BC | National Center for Environmental Prediction Final Analysis (NCEP-FNL) |
| Chemical IC and BC | European Centre for Medium-Range Weather Forecasts (ECMWF) |
| Emissions sources | |
| Biogenic | Model of Emissions of Gases and Aerosols from Nature (Guenther et al., 2006) |
| Anthropogenic | Emission inventory developed by Rafee et al. (2017) |

plant functional types, this model estimates the net terrestrial biosphere emission rates for different trace gases and aerosols with a global coverage of $\approx 1$ km$^2$ spatial resolution.

## 2.3 Observed Data

We used in situ real-time measurement at several GoAmazon2014/5 surface sites (see Fig. 1b). The particle scattering coef-

ficient ($\sigma_s$) was measured using a 3-wavelength Nephelometer (450, 550 and 700nm; TSI 3563 Integrating Nephelometer). Particle absorption coefficient ($\sigma_a$) was measured at the T3 site with a 7-wavelength Magee AE31 Aethalometer that operates at $\lambda = 370, 430, 470, 520, 565, 700$ and 880 nm and was subjected to the correction scheme outlined by Rizzo et al. (2011). The observed ($\sigma_a$) values have been interpolated to the nephelometer's wavelengths to allow a proper comparison and calculation of the intensive parameters, such as SSA. The BC mass concentration at the T3 site was estimated using AE31 measurements of





the absorption coefficient at 880 nm and a mass absorption cross section (MAC) section of 7.77 $m^2 g^{-1}$ (Drinovec et al., 2015). At ATTO, the BC concentration was measured using a Thermo Environment MAAP 5012 (Thermo) using a $\sigma_a$ at 637nm and a MAC of 6.6 $m^2 g^{-1}$, the absorption data was corrected according to Müller et al. (2011). Organic and inorganic submicron aerosol mass loadings were measured with a Time of Flight Aerosol Mass Spectrometer (ToF-AMS) (de Sá et al., 2018). Mixing rations of ozone ($O_3$) and CO were obtained with a 49i $O_3$ Analyzer (Thermo Environment) and a $N_2O$/CO Analyzer (Los

Gatos Research - LGR). Meteorological observations was measured by a Vaisala WXT520. Observed data was averaged at 1-hour intervals for comparison it with the WRF. Standard temperature and pressure (STP) corrections were also applied to all measurements. We also used aircraft measurements of $\sigma_a$ from the DoE Gulfstream 1 (G-1), as part of the GoAmazon2014/5 experiment (Shilling et al., 2018; Martin et al., 2016), measured using a 3-wavelength (461, 522 and 648nm) Particle/Soot Absorption Photometer (PSAP) from Radiance Research.

### 2.3.1 GoAmazon2014/5 Experiment

The Observations and Modeling of the Green Ocean Amazon experiment GoAmazon2014/5 was designed to understand how aerosol and cloud life cycles are influenced by the pollutant outflow from Manaus into the tropical rain forest (Martin et al., 2016). The experiment used a set of detailed aerosol, trace gas and cloud measurements at six different sites (see Fig. 1b) in order to better understand the atmospheric processes caused by the interaction between urban pollution emissions with volatile

organic compounds (VOCs) emitted from the forest, and the environmental impacts on the natural microphysical properties of clouds and aerosols, such as optical properties and particle size distributions (Gu et al., 2017; Fraund et al., 2017).

## 3 Results and Discussion

### 3.1 Meteorological Analysis

To study the impact that Manaus pollution plume has on SOA production aerosol optical properties in the area downwind

of Manaus, meteorological conditions, especially temperature, humidity and PBL height, must be properly characterized and represented in the WRF-Chem model. Comparisons at the T3 site between observed and simulated hourly variations of accumulated total precipitation, 2 m temperature, 2 m relative humidity, 10 m wind speed, and PBL height (SI Figs. S1 and S2) show that the model performs well in terms of diurnal representation and trends. Simulated temperature and humidity tend to be underestimated (mean bias (MB) = -0.5 and -1.6, respectively), with a short delay between peak observed (11:00 LT) and

simulated (15:00 LT) values. The simulation has difficulties in obtaining the observed maximum temperature (see SI Fig. S1a). According to statistical indices (SI Table 1), the correlation coefficient (*r*) and Root Mean Square Error (RMSE) show consistent results for relative humidity (*r* = 0.7 and RMSE = 1.8), temperature (*r* = 0.8 and RMSE = 0.4) and wind speed (*r* = 0.7 and RMSE = 0.2). The relative humidity profile agrees well with ground base measurements, but the simulated values exhibit the diurnal minimum with a 3 hour delay. Individual calculations of performance statistics are presented in Supplementary Table

S1.





The Central Amazon region has unique topographic characteristics including the Amazon, Negro and Solimões rivers (Marinho et al., 2020), resulting in meteorological systems such as local circulations and the so called *friagem* events, which occur when a frontal system reaches the Central Amazon basin (Marengo et al., 1997; Lu et al., 2005), that have important influences on the local and mesoscale circulations (dos Santos et al., 2014; Pereira Oliveira and Fitzjarrald, 1993; Silva Dias et al., 2004).

That may affect the wind direction and air subsidence patterns. Figure S3 in the SI compares the simulated vertical wind component during night time at the T3 site. In the early morning hours (05 - 11 LT), downdraft movement is not sufficient at the T3 site to inhibit pollutant dispersion. However, during the night time (20 - 22 LT), the simulation captured an organic aerosol concentration peak (see Fig. 5a) consistent with the presence of downdraft movement and a temperature inversion at low levels (see SI Fig. S4) observed at the T3 site.

### 3.1.1 Background Conditions

Generally, global and regional models contain uncertainties associated with the wet/dry deposition scheme (Wang et al., 2015). For example, the BC residence time in the atmosphere is typically larger in global models than in the real atmosphere. During the wet season, the T0a site is upwind of Manaus and so has low anthropogenic influence. However, the T0a site receives sporadic air masses loaded with marine aerosol transported from the Atlantic Ocean, and dust outflows from the Sahara desert,

in general together with smoke from fires in West Africa (Ben-Ami et al., 2010; Andreae et al., 2012, 2015; Rizzolo et al., 2017; Pöhlker et al., 2018). Those air masses transported from Africa during the wet season occur when the ITCZ is shifted to the south of the central Amazonian Basin, allowing air masses from the Northern Hemisphere to reach the central portion of the Basin.

Looking at Figure 2, it is possible to observe that on March 2014 $10^{th}$ and $11^{th}$, BC (both simulated and observed) were

220 above expected levels (0.1 $\mu gm^{-3}$), consistent with coherent BC long range transport from west Africa (Moran-Zuloaga et al., 2018). During the $10^{th}$ and the $11^{th}$, the simulation also follows the BC variability shown in the observed data. The simulation appears to do a reasonable job of representing the BC transport from West Africa. Figure 2 shows that the global model BC concentrations are also representative of the hours with the largest values during the $10^{th}$ and $11^{th}$.

March $13^{th}$ shows good agreement between simulated and observed data at T3. We can assess the simulations ability to

225 represent Amazonian background conditions comparing observed and simulated data from the region with very little anthropogenic influence during the wet season, as is the case for the T0a site. Given that, the simulation calculates background BC and $O_3$ with RMSE and MB consistent with observations, it is possible to determine the aerosol enrichment at the T3 site due to the Manaus plume. The background BC in the Amazon near the T0a site is representative of almost pristine continental regions. The average observed BC values are influenced by biogenic aerosol absorption, the global BC background as well

as by BC that is long range transported from Saharan dust and African biomass burning. The BC transported from Africa is episodic, depending on the ITCZ positioning, as well as the air mass trajectories from Africa to the Central Amazon. As we have several years of BC background measurements at the ATTO tower, it is possible to separate African episodic events from the rather constant regional BC concentrations that are relevant to compare with the modeled values under no anthropogenic influences (Artaxo et al., 2020).





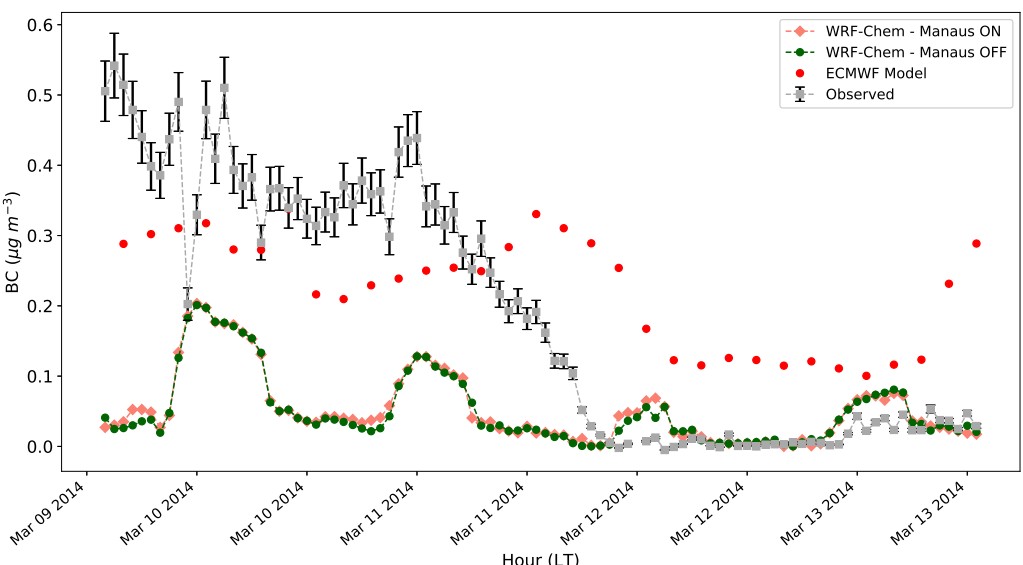

**Figure 2.** Observed and simulated surface BC concentration from March 10 to March 13, 2014 at the T0a site. Standard deviation bars are shown for each set of measurements. Events due to long range transport of Saharan dust and biomass burning emissions from West Africa are visible on the $10^{th}$ and $11^{th}$ (Moran-Zuloaga et al., 2018). The Manaus OFF (green) and Manaus ON (orange) simulations show BC concentrations simulated at a height of ca. 8 m above the surface. During BC transport event days, we can see that the simulation had the largest peaks, trying to represent the black carbon transport coming from west Africa. The global model contribution (red dots) also represents BC transport event days, showing the largest values during days $10^{th}$ and $11^{th}$.

## 3.2 Chemical Analyses

To better understand the impact of the Manaus urban plume on SOA formation and mixing ratios at the T3 site during March 13, 2014 we must be able to separate time periods representing clean and polluted episodes, and compare observed and simulated values. Previous studies have developed methods to separate these episodes in the Amazon region (Palm et al., 2017; de Sá et al., 2018; Cirino et al., 2018).

In our analysis, with observed data from the GoAmazon2014/5 experiment (T3 site), adjusted cluster centroids were used to analyze clean and polluted conditions, during two months in the wet season (February and March 2014). We define three different clusters (i) low pollution (Low Pol), (ii) middle pollution (Mid Pol) and (iii) high pollution (High Pol) (see Table 2). We chose three different clusters at the T3 site because the pollution conditions arriving are heterogeneous. Our cluster analysis (see Fig. 3) was made with a fuzzy c-means (FCM) clustering algorithm (Bezdek et al., 1984). On March 13, 2014, our analysis shows a day with mostly polluted conditions (at 10-17 LT). Previous work (Palm et al., 2017; de Sá et al., 2018) reported the same polluted conditions during this day.





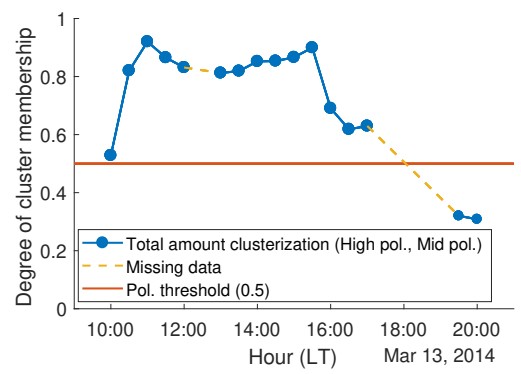

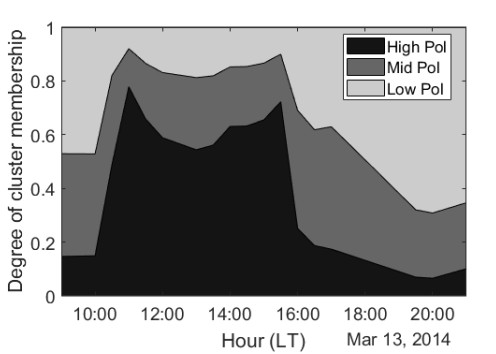

**Figure 3.** Results of FCM clusters analysis during March 13, 2014 from 10:00 to 20:00 LT. **(a)** Total clusterization considering polluted conditions with degree of cluster membership $> 0.5$. **(b)** Degree of membership in each of the three clusters. The sum of degrees of membership across all clusters is unity. Background conditions are abbreviated as "Low Pol", intermediate conditions as "Mid Pol" and polluted conditions are abbreviated as "High Pol".

**Table 2.** Cluster centroids used to analyze clean and polluted conditions.

| Cluster Centroids for March Day 13th | | | | | | |
|---|---|---|---|---|---|---|
| Clusters | $PM_{2.5}$ conc. num. $(cm^{-3})$ | CO (ppbv) | $O_3$ (ppbv) | BC $(ng/m^{-3})$ | $NO_y$ (ppbv) | $SO_4$ $(\mu g/m^{-3})$ |
| Low Pol. | 1304 | 117 | 11 | 43 | 0.71 | 0.16 |
| Mid. Pol. | 2566 | 123 | 15 | 99 | 1.39 | 0.29 |
| High Pol. | 5329 | 124 | 26 | 144 | 2.28 | 0.43 |

Because the concentration values of High Pol and Mid Pol, episodes are substantially larger than those at Low Pol we distinguish time periods representing clean episodes as Low Pol and polluted episodes as High Pol and Mid Pol. Quantitatively we separate clean from polluted episodes with the degree of cluster membership. When membership for Low Pol is $> 0.5$,

we consider this a clean episode. When the sum of Mid Pol and High Pol membership is $> 0.5$ we consider this a polluted episode. Initially, we attempted clustering with only two clusters (one for clean and one for polluted episodes), but were unable to separate polluted from background conditions. In this case the nominally background cluster had high black carbon (BC) and $NO_y$ concentrations.

Given the abundance of BVOCs in the Amazon region (Alves et al., 2016; Yáñez-Serrano et al., 2020), we expect $O_3$ to be

especially sensitive to changes in $NO_x$ emission. This can be seen in Figure 4a; 4e, which shows high $O_3$ and low $NO_x$ values downwind of Manaus. According to the WRF-Chem chemical mechanism, isoprene is rapidly oxidized by hydroxyl radicals (OH) to form peroxy radicals ($HO_2$) in a few hours (Ahmadov et al., 2012). The T1 site has a low isoprene concentration, since as the Manaus plume passes through forest regions with high isoprene production, the high plume $NO_x$ concentration oxidizes the isoprene. This can be seen in Figure 4, where the Manaus plume consumes the isoprene around the T3 site, producing $O_3$





and $HO_2$. Because the enhancement of $HO_2$ radicals occurs downwind of Manaus (such as at the T3 site), the concentrations of $NO_x$ are significantly lower (higher) than the values in Manaus (downwind in the forest), leading to a significant enhancement of $O_3$ (ca. 8 – 30 pppv (Fig. 4a). Because $NO_x$ and isoprene emissions vary in different regions, our results suggest that $NO_x$ in southeastern Manaus (Rafee et al., 2017) has important impacts on the $O_3$ concentration in the Manaus urban area. This is primarily due to the rapid reactions of radicals with $NO_x$, which deplete the radicals.

The $O_3$ values are highest during the day as VOC production peaks and solar radiation is available for the photo-chemical processes that produce $O_3$ (Graham et al., 2003a, b; Chen et al., 2015; Schultz et al., 2017). The $O_3$ enhancement 8 to 300 km downwind of Manaus suggests that the interaction between forest biogenic emissions and the pollution from Manaus could have an important impact on the chemical production of $O_3$ (Fig. 4a). The interaction between anthropogenic and biogenic trace gases has strong regional characteristics, such the ones found near Manaus. They also depend on the distributions of BVOCs and anthropogenic $NO_x$. $O_3$ mixing ratios downwind of Manaus under the influence of anthropogenic pollution were also reported by Trebs et al. (2012) and were on average $31 \pm 14$ ppbv, with peak values of 60 ppbv at a distance of 19 km downwind of Manaus. Our simulations showed an $O_3$ average of $30 \pm 11$ ppbv at the T3 site (70 km downwind of Manaus) with high peak values of 148 ppbv in regions northwest of Manaus (Fig. 4a). Manaus pollution plume influence on $O_3$ production is clearly observed in the surrounding area, predominantly to the west and northwest of Manaus.

In regions downwind of Manaus, the simulations showed $O_3$ concentrations extending more than 300 km downwind. It is also interesting to note the lower $O_3$ values around T1, which are represented in both observed (ca. 8 ppbv on average) and simulated (ca. 12 ppbv on average) data (Fig. 4a). $O_3$ with ca. 8 ppbv on average is uncommonly low for a metropolis of nearly 1.8 million people. The agreement between observed and simulated $O_3$ values around T1 indicates that the chemistry there is being successfully reproduced by the simulation. Our explanation for this anomaly is that VOCs are abundant all around Manaus (Kuhn et al., 2010; Alves et al., 2016) and $HO_x$ and $O_3$ are low despite having high $NO_x$ in a typically VOC limited regime. We hypothesize that in areas with very high $NO_x$ emissions (averaging 129.02 ppbv), such as the power plant cluster surrounding T1 (Fig. 4e), radicals react quickly with $NO_x$ ($NO_x + OH \rightarrow HNO_3$). This depletes the $O_3$, creating radicals, causing a decrease in $O_3$ formation. Conversely, downwind of Manaus, the radicals last long enough to form $O_3$ and we observe an increase in $O_3$ formation, as well as an increase in $HO_2$ radicals (Fig. 4c).

Our results imply that the high $NO_x$ conditions within Manaus affect the $O_3$ formation around Manaus, decreasing $O_3$ production with in the city and providing a great enhancement downwind of Manaus (Fig. 4a). The wind direction is predominantly from the northeast, which allows the plume be transported to the T2 and T3 sites and allowing the pollution plume to have a great impact on the surrounding areas (Martin et al., 2017). Interestingly, our results show that when $O_3$ concentrations change by a factor of between 2 and 4, oxidation and $NO_x$ levels may be affected, and consequently, the rate and SOA production efficiency may be impacted, by decreasing the reacted BVOCs and SOA formation downwind of Manaus.

According to Figure 5a, the simulated organic $PM_{2.5}$ at the T3 site has one of the highest values during the first hours of March 13, 2014 (2 to 4, LT), with the largest contribution coming from primary anthropogenic organic aerosol (POA). We suggest that the large contributions of BC and CO emissions, coming from Manaus (Fig. S7 in the SI) together with a prevailing northeast wind direction, are the most plausible explanations why simulated total organics present high values





**Figure 4.** Temporal mean (06 to 15 LT March 13th) spatial distribution of simulated surface level concentrations of (a) $O_3$, (b) Isoprene, (c) $HO_2$, (d) OH, (e) $NO_x$ and (f) daily median $O_3$ profile for the month of March (wet season) at T1 during 2014 (green line), 2016 (black line) and at the T3 site during 2014 (red line). The red, gray and green shaded areas show the $25_{th}$ to $75_{th}$ percentiles of the respective median line.





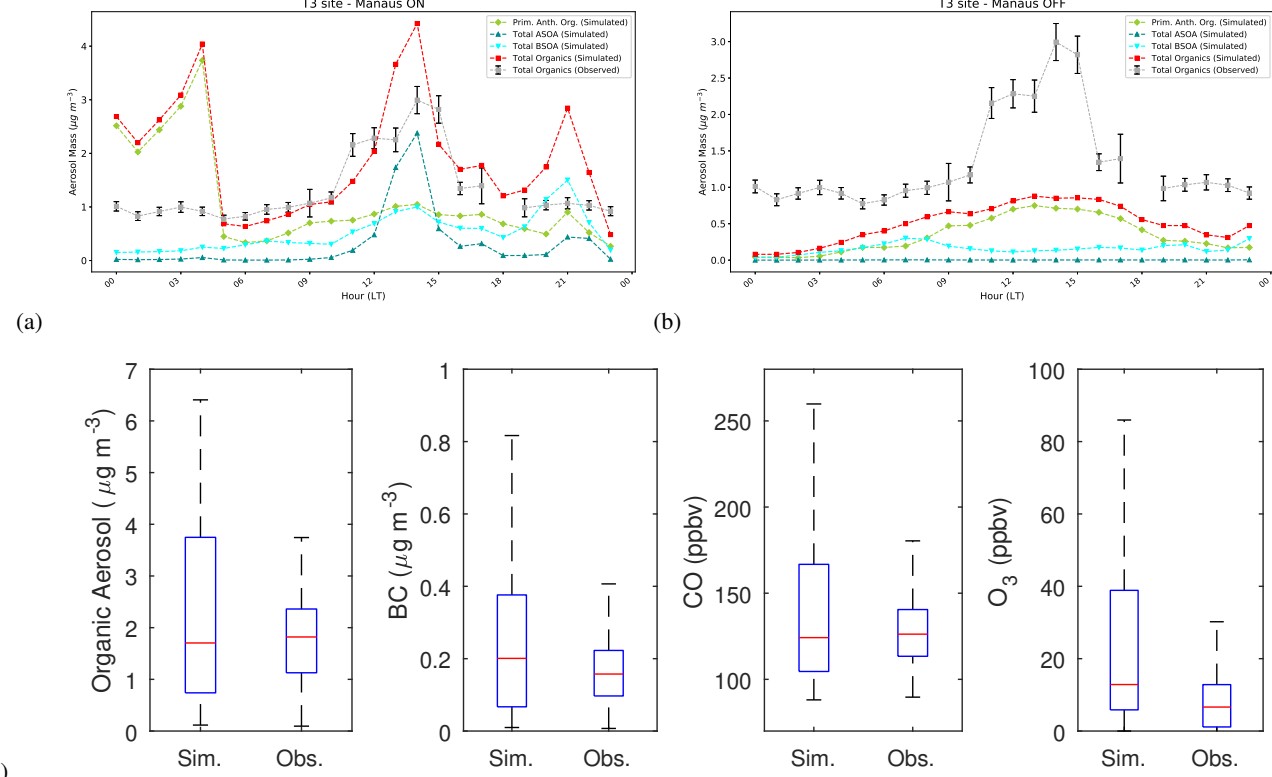

(a)  (b)

(c)

**Figure 5.** Times series and box plot comparison of measured and WRF-Chem-simulated surface level gases and aerosols at the T3 site. Contributions from simulated primary anthropogenic organic, BSOA and ASOA to total OA as simulated by WRF-Chem in the case (a) with and (b) without anthropogenic emissions on March 13, 2014 at the T3 site. (c) Comparison between observed and simulated surface level gases and aerosols. Box plot of simulated and observed organics, BC, CO and $O_3$ from March 9 to 13, 2014 at the T3 site. Median values are shown in red lines and the blue box indicates data between the $25^{th}$ and $75^{th}$ percentiles.

during the first hours of the day with an high POA peak during the first hours of March 13, 2014 (2 to 4, LT). The BC and CO contributions can end up reaching the T3 site, increasing the POA amount. In addition, the simulated BC concentration also showed simultaneously high values during the first hours of March 13, 2014 (2 to 4, LT) (Fig. S5 in the SI).

Between 10 and 16 LT there is an increase in the total organic aerosol concentration, which was successfully reproduced by our simulation. This evolution of the organic aerosol concentration was expected on that day due to the Manaus plume arriving

at the T3 site (Shilling et al., 2018). This increase is mostly due to a sharp increase in anthropogenic SOA (ASOA) peaking at 15 LT, as well as the BSOA and POA at the same time. The highest value (4.4 $\mu g\ m^{-3}$) of simulated total organics occurred at 14 LT (Fig. 5a), and is comprised of mostly the SOA component, with increases in BSOA contributing 1.0 $\mu g\ m^{-3}$ (22.6%), ASOA 2.4 $\mu g\ m^{-3}$ (53.9%) and POA 1.04 $\mu g\ m^{-3}$ (23.5%). Conversely, when the simulation is run with anthropogenic emissions turned off, the total organic aerosol simulated at 14 LT is 0.9 $\mu g\ m^{-3}$, with BSOA contributing 0.14 $\mu g\ m^{-3}$ (16.3%), ASOA





0.02 $\mu g\ m^{-3}$ (2.3%) and POA 0.7 $\mu g\ m^{-3}$ (81.4%). We attribute most of the difference in total organic aerosol between simulations with and without anthropogenic emissions to the ASOA amount, related the Manaus plume contributions. The same analysis, now considering the entire day of March 13th, shows a contribution coming mostly from POA of 26.4 $\mu g\ m^{-3}$ (57.1%), BSOA 12.4 $\mu g\ m^{-3}$ (26.8%) and ASOA 7.4 $\mu g\ m^{-3}$ (16%). Considering the immensely complex mixture of organic aerosol particle and gas phase, VOCs and other species in continuous evolution in the atmosphere, and the large number of

chemical reactions with oxidant species such as OH (day-time) and $NO_3$ (nighttime) (Kuhn et al., 2010), we emphasize that there may be a relationship between BSOA and ASOA simulated peaks (see Fig. 5a)) and the $O_3$ peak at 15 LT (Fig. S6 in the SI), since those chemical reactions are associated with the production of tropospheric $O_3$ and also oxygenated VOCs (Bela et al., 2016).

A third total organic aerosol simulated peak is observed between 20 and 21 LT (see Fig. 5a). The simulated peak may be

explained by the transport of air pollutants from the regions south of the T3 site (Fig. S8 in the SI). We propose two possible explanations for this phenomenon. Our first explanation involves the Negro River breeze effect. Since most thermal power plants and the Issac Sabbá refiner REMAN are located near the banks of the Negro and Solimões rivers (Rafee et al., 2017), the plume transport could be influenced by the river breeze circulation, which defines the trajectory of pollutants. It may be that, between 19 to 21 LT (Fig. S8 in the SI), the wind direction was affected by the Negro River breeze effect due the horizontal

thermal gradient caused by the different energy partitioning of the water and land surfaces. Consistent with dos Santos et al. (2014), the water surface temperature of the Negro River starts to increase in the afternoon (13 LT), affecting the vertical heat and mass transport. Our second explanation is that there is an air subsidence pattern at the T3 site between 19 and 22 LT (see SI Fig. S3). At 20 LT T3 site presents a saturation trend from 850 m to 900 m and also from 520 m to 600 m with temperature and dew-point temperature close to each other, creating a dry air region (see Fig. S4 in the SI), and consequently, air subsidence

(see Fig. S4b in the SI). This causes upward movement inhibition, which confines the atmospheric pollutants to low levels, impeding their spread.

An example of the differences between the measured and modeled concentration distributions is shown for organics, BC, CO and $O_3$ in Fig. 5c. Both simulated and observed BC show a median value of 0.2 $\mu g\ m^{-3}$, demonstrating that our simulation represents BC well. The same behavior is shown for OA and CO with simulated and observed median values of 1.8 $\mu g\ m^{-3}$ and

122 ppbv, respectively. However, the simulation presets a larger range of values compared with observations. The simulations present some high peaks not seen in the observed data, such as the ones in BC (see Fig. S5 in the SI) and OA, with a high contribution coming from POA emission factor (see Fig. 5a). Both have peaks in the early morning on March 13, 2014. In addition, the simulation shows a median $O_3$ value of ca. 12 ppbv (observed 7 ppbv). Conversely, looking at just 10 to 17 LT on March 2014, which represents an exemplary day with pollution contributions at the T3 site coming from Manaus (Shilling

et al., 2018), both simulated and observed $O_3$ present high median values, 38 ppbv and 30 ppbv, respectively. This agreement of the observed and simulated median values during a day with polluted conditions, particularly noting the uncertainties in emissions (speciation, spatial and temporal distribution), measurements, boundary conditions, meteorological component and other input parameterization of the model are low on the $13^{th}$. Overall, the comparisons of the median measured and predicted





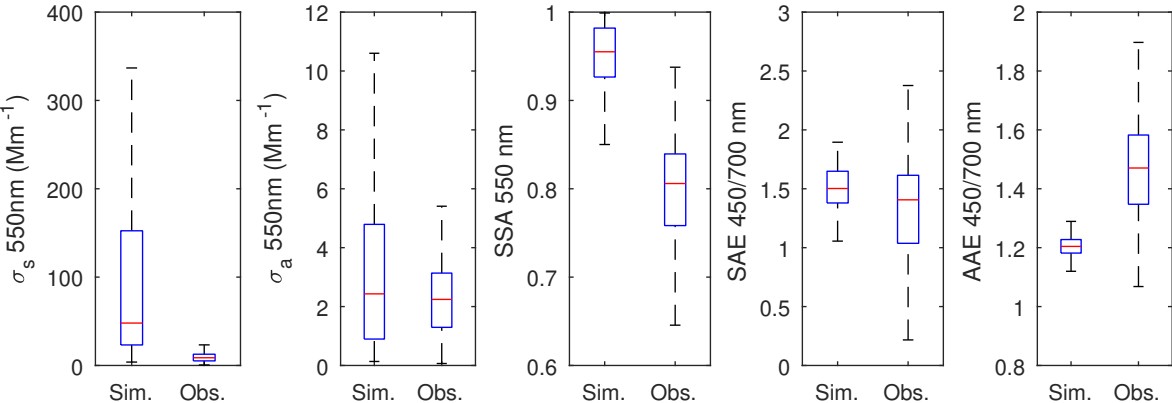

**Figure 6.** Comparisons between observed and simulated aerosol optical properties. Box plot of simulated and observed SSA, SAE, AAE, scattering and absorption coefficients from 9 to 13 March 2014 at the T3 site. Median values are shown in red lines and the blue box indicates data between the $25^{th}$ and $75^{th}$ percentiles.

chemical concentrations are satisfactory, with the best match obtained in OA (observed 1.8 $\mu g\ m^{-3}$; simulated 1.7 $\mu g\ m^{-3}$),

CO (observed 124 ppbv; simulated 126 ppbv) and BC (observed 0.201 $\mu g\ m^{-3}$; simulated 0.203 $\mu g\ m^{-3}$).

### 3.3 Variability of Amazonian Aerosol Optical Properties

Understanding how optical properties such as SSA and $g_{aer}$ vary downwind of Manaus is key to understanding the impact of the pollution plume on radiative forcing, its contributions to the local radiative budget, its impacts on the hydrological cycle and unknown indirect consequences on photosynthesis rates. These effects suggest the possibility of investigating aerosol

direct radiative effects (DREs) by examining $g_{aer}$, which presents, in general, higher values associated with stronger forward scattering of radiation by atmospheric aerosols (Korras-Carraca et al., 2015).

 Figure 6 shows that the simulation overestimates the observed scattering coefficient by a factor 6. The overestimate in the observed scattering coefficient is due the fact that our WRF-Chem simulations are producing more $SO_4$ than in the real atmosphere, with 30% of the observed PM1 attributed to $SO_4$ in the accumulation mode (Fig. S11 in the SI). Observed scattering

coefficient values are significantly lower than simulated likely due to decreases in the aerosol loading during the transect, modulated by the effects of dilution of gases and particles in the air. On the other hand, the median simulated absorption coefficient of 2.2 Mm$^{-1}$ is in good agreement with the observed median value of 2.4 Mm$^{-1}$. We observe the simulated SSA being affected by the simulated scattering coefficient overestimation. Comparing simulated and observed SAE values, we again have good agreement between the simulation and observations, with the simulation representing the mean size of the aerosol population

70 km downwind of Manaus quite well. These results are important for the plume aging mechanism discussed in Section 3.4. Additionally, the observed AAE is considerably higher than in our simulation. This suggests that the brown carbon component,

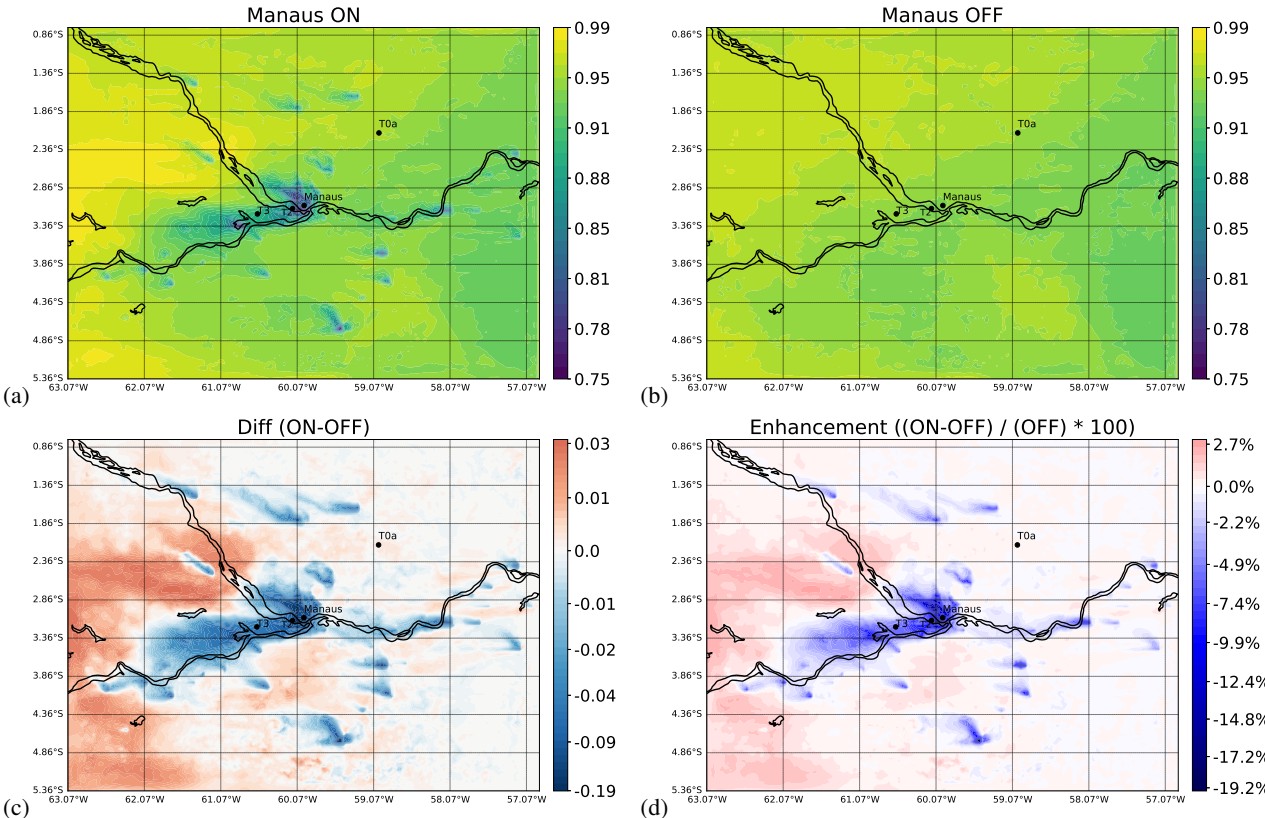

**Figure 7.** WRF-Chem simulated values of SSA in the presence or absence of Manaus emissions. (a) SSA when all emissions are ON. (b) SSA when just biogenic emissions are ON and anthropogenic emissions are OFF. (c) SSA difference between the two simulations with anthropogenic emissions turned ON and OFF i.e. (ON-OFF). (d) SSA enhancement (%) calculated from the two simulations with anthropogenic emissions turned ON/OFF i.e. ((ON–OFF)/OFF)×100. WRF-Chem predictions are at ca. 8 m altitude, averaged over March 13, 2014 (00:00 – 23:00 LT.))

not accounted for our simulations, could have an critical effect on the AAE value, contributing to the lower median simulated AAE (1.2) compared with the median observed value (1.5).

### 3.3.1 Calculations and measurements of SSA

According to our simulation results, the Manaus plume interferes with the amount of radiation absorbed by the atmosphere, being responsible for an SSA reduction of approximately 10% at Manaus, 12% at the T2 site and 5.3% at the T3 site (see Fig. 7d). This indicates a large fraction of absorbing material present in the Manaus plume, potentially warming the local atmosphere. These regions are associated with thermal power plants (Medeiros et al., 2017), indicating that the vehicular emissions and stationary sources (refineries) are dominated by small absorbing particles like BC, while biogenic particles are

mostly found in the coarse mode and efficiently scatter radiation due to their organic carbon-dominated composition.





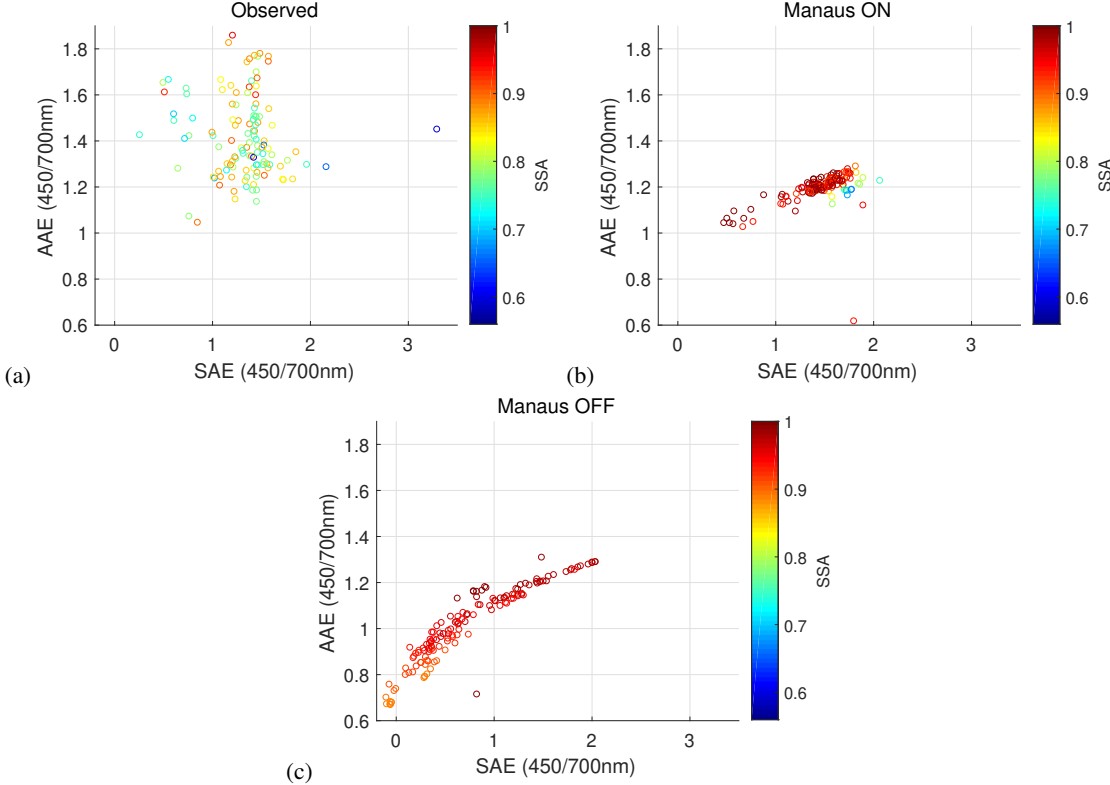

**Figure 8.** AAEs at the wavelength pair 470 nm and 660 nm as a function of the corresponding SAEs at the wavelength pair 470 nm and 660 nm), color-coded using the related SSA at the wavelength pair 470 nm and 660 nm. 1 h averaged instantaneous observed data values (a) from simultaneous nephelometer and aethalometer measurements from 9 to 14 March 2014. (b) 1 h averaged simulated values when all emissions are on. (b) 1 h average simulated values when just biogenic emissions are on and anthropogenic emissions are off.

During simulations with the Manaus pollution plume component turned on, average SSA values vary between 0.75 and 0.90 in regions downwind of Manaus. This represents the contribution from the interaction of urban aerosols with biogenic components of the forest. Similar results were found by Cirino et al. (2018) at the T3 site (0.80) and Ramachandran and Rajesh (2007) in western India (0.88), He et al. in China (0.80), Backman et al. (2012) in São Paulo, Brazil (0.76). These SSA values

are associated with the formation of SOA aerosols which scatter radiation efficiently (Fig. S10 in the SI). The decrease in the SSA is associated with a significant fraction of aerosol loading from small particles of anthropogenic origin, e.g., BC. The average simulated and observed SSA on 550 nm values during March 13, 2014 at the T3 site were $0.86 \pm 0.09$ and $0.78 \pm 0.09$, respectively.



### 3.3.2 Calculations of AAE and SAE

Figure 8 shows the simulated and observed SAE and AAE distributions from 9 to 14 March 2014. The simulation with anthropogenic emissions is mostly characterized by 1.0 < AAE < 1.3 and 1.0 < SAE < 2.0, corresponding to a large OC particle contribution, including primary and secondary components (POC and SOC, respectively) (Cazorla et al., 2013). Additionally, the simulated SAE (Manaus on) when variability ranges between 1 to 1.8, indicates a contribution of fine and absorbing particles, which increases the SAE (see Fig. 8).

In general, these SAE and AAE values show that the values in simulation with anthropogenic emissions are, on average, associated with the fine fraction of $PM_{2.5}$ sampled particles. In contrast, some values are mostly associated with large-size $PM_{2.5}$ particles (SAE < 1), consistent with the Manaus plume not having a strong influence on the T3 site during those days. Conversely, the SAE with anthropogenic emissions (see Fig. 8b) shows a range between 0.5 and 2.1, values associated with the presence of fine aerosols originating from industrial activities in Manaus and the thermal power plants (TTPs) located in

the surrounding area. The simulation with the Manaus plume turned off (see Fig. 8c) shows a coarse mode predominance, with SAE values varying mainly between 0.0 and 1.5. Thus, we can assume those values have a large OC contribution because of the predominance of aerosol coming from coarse mode biogenic sources.

The observed AAE values in the simulation without anthropogenic emissions express a large variability (1.1 to 1.8) compared with the ones from simulation with anthropogenic emissions (1 to 1.3). This behavior is assumed being caused by the lack of

390 a brown carbon component in the aerosol population in our simulation. When the anthropogenic emissions are off, the SAE variability is mostly related with the significant contribution from large aerosol, as already mentioned (Cazorla et al., 2013; Seinfeld and Pandis, 2016; Romano et al., 2019).

### 3.3.3 Asymmetry Parameter

$G_{aer}$ is an important optical property in radiative transfer, climate and general circulation models (Korras-Carraca et al. (2015).

The, $g_{aer}$ describes the angular distribution of scattered radiation and determines whether the particles scatter radiation preferentially forwards or backwards (Boucher (2015)).

Figure 9 (a) shows low 600nm $g_{aer}$ values (0.64) that could be associated with industrial activities such as TTPs as well as biomass burning in nearby areas. A region of special interest is between Manaus and T3, since it hosts a large variety of mixing interactions between anthropogenic, biogenic and dust aerosols (e.g., Artaxo et al., 2002; Saturno et al., 2018; Martin et al.,

2016; Rizzo et al., 2013). In this region it can be seen that $g_{aer}$ decreases by 8% compared when there is no anthropogenic emissions (see Fig. 9d). This is associated with the presence of fine anthropogenic aerosols transported from adjacent urban and industrial areas in the northwest, especially from central Manaus (Medeiros et al., 2017; Rafee et al., 2017; Shrivastava et al., 2019). Those smaller $g_{aer}$ values are seen in places where a significant fraction of the aerosol loading comes from small size particles of anthropogenic origin, with the smallest values appearing over the regions containing industrial activities.

Previous studies (Cirino et al., 2018) have shown a period in the late afternoon around T3 in which particles with the smallest geometric diameter (ca. 50 nm) were observed, and the same period coincides with smaller $g_{aer}$ found in simulations for the





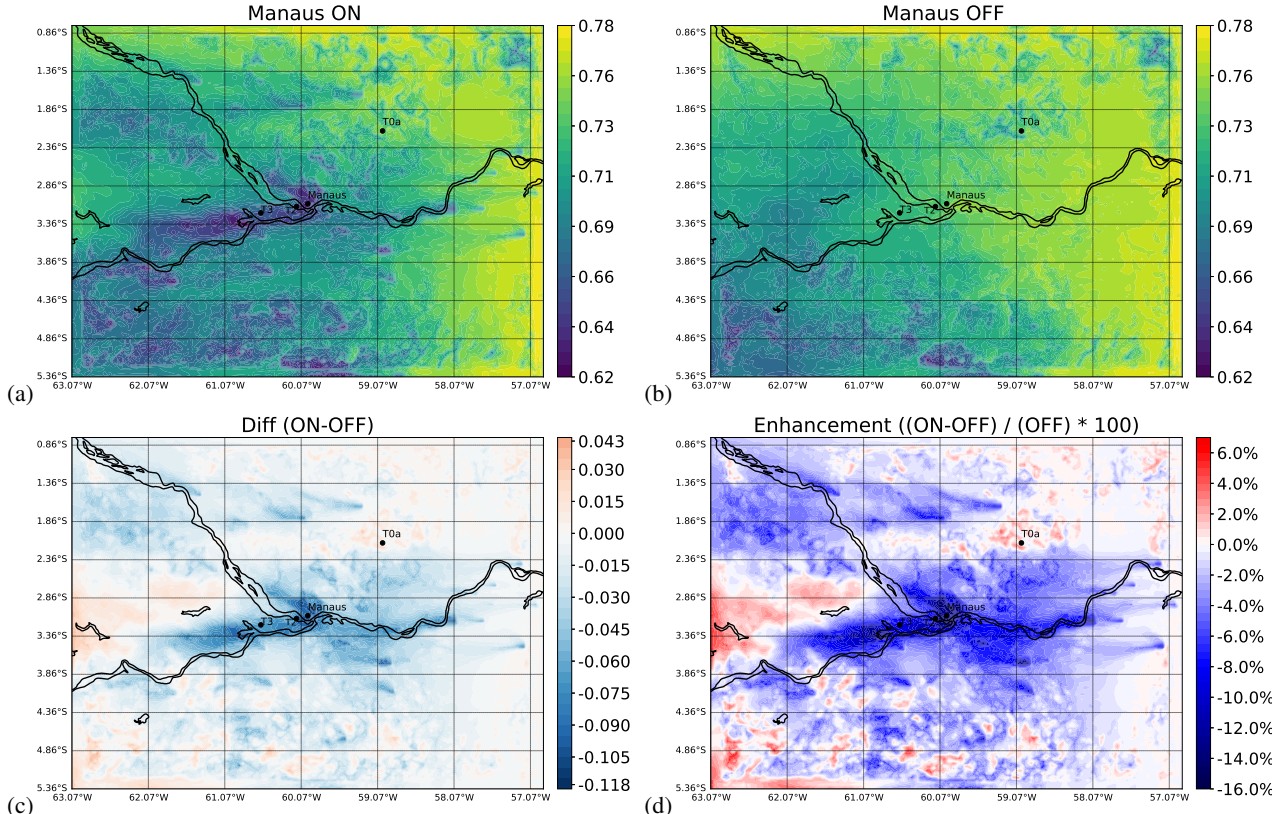

**Figure 9.** WRF-Chem simulated values of 600nm $g_{aer}$ in the presence or absence of Manaus emissions. (a) $g_{aer}$ when all emissions are on. (b) $g_{aer}$ when just biogenic emissions are on and anthropogenic emissions are off. (c) $g_{aer}$ difference between the two simulations with anthropogenic emissions turned on and off i.e. (ON-OFF). (d) $g_{aer}$ (%) enhancement calculated from the two simulations when anthropogenic emissions turned on and off i.e. ((ON–OFF)/OFF)×100. WRF-Chem predictions are at ca. 8 m altitude, averaged over March 13, 2014 (0 to 23 LT).

same station (see Fig. 9a). On the other hand, $g_{aer}$ when anthropogenic emissions are off, $g_{aer}$ has predominately large values varying between 0.75 and 0.76 at 300 nm, 0.73 and 0.75 at 400 nm, 0.71 and 0.74 at 600 nm and 0.63 and 0.71 at 1000 nm. These values indicate strong forward scattering of radiation by atmospheric aerosols and are related with the presence of coarse biogenic particles. According to the obtained results, anthropogenic emissions decrease $g_{aer}$ values by between 2% and 16%, especially in regions with large mobile and stationary anthropogenic activities. Those smaller values can induce modifications of the DREs.

### 3.3.4 Irradiance

In regions like the Amazon with sufficiently high levels of $NO_x$, and VOCs such as isoprene and monoterpene, an enhanced formation of near surface $O_3$ is expected. Solar radiation is another element that contributes to photochemical activity and,





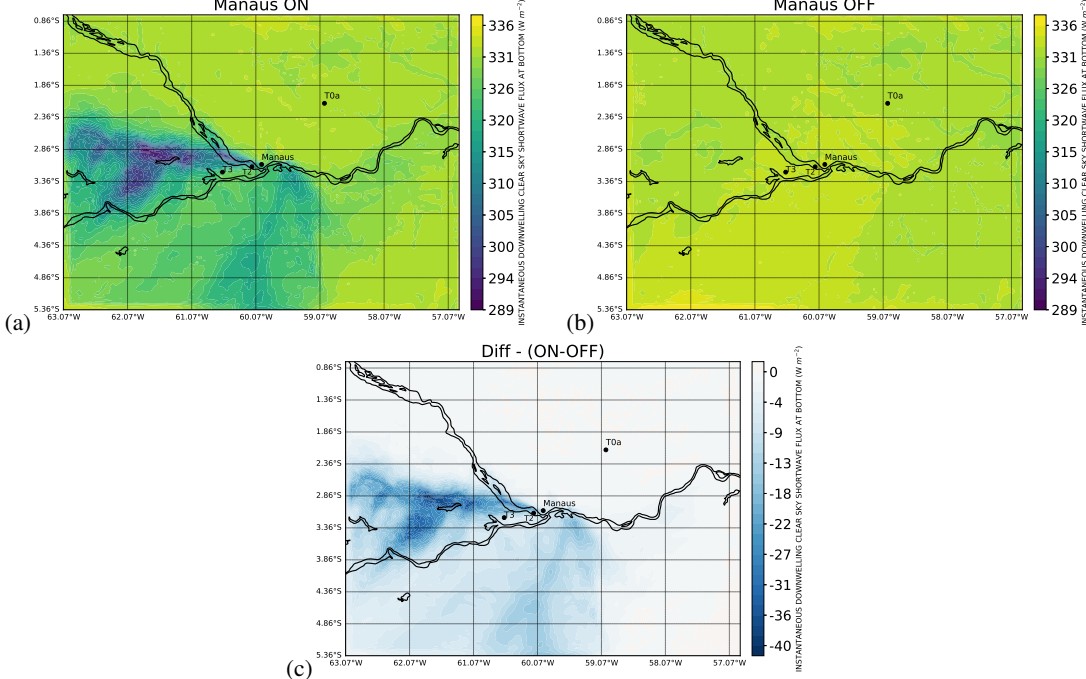

**Figure 10.** WRF-Chem simulated mean incoming solar radiation (instantaneous downwelling clear sky shortwave flux at bottom: SWDNBC) in $Wm^{-2}$ in the presence and absence of Manaus emissions. (a) SWDNBC when all emissions are on. (b) SWDNBC when biogenic emissions are on and anthropogenic emissions are off. (c) SWDNBC difference between the simulations with and without anthropogenic emissions, i.e. ON-OFF.

consequently, the formation of $O_3$. According to Figure 10c, it is possible to notice that even in regions presenting average decreased surface downward shortwave flux values of ca. 20 W m$^{-2}$ due to the presence of anthropogenic emissions near T2 and T3, not enough to reduce the enhanced formation of near surface $O_3$ (see Fig. 4a), which more than compensates for the effect of the comparatively reduced solar radiation there. The lower solar radiation over the west side of Manaus seen in

simulations with anthropogenic emissions (see Fig. 10a) is accompanied by a general increase in mean $O_3$ values (see Fig. 4a). Studies of regional direct and indirect aerosol effects are important and still challenging due to their complexity making an accurate determination of the direct and indirect effects difficult (Forkel et al., 2012; Wang et al., 2015; Zhang et al., 2010).

     In our simulations, we considered both direct and indirect aerosol effects during the wet season in the Amazon region. Incoming shortwave radiation at the surface is predicted to drop by up to ca. -40 W m$^{-2}$ due the direct aerosol effect. In regions

within and up to ca. 100 km south-west of Manaus, Figure 10c shows an aerosol cooling effect with maximum SWDNBC values ca. -40 W m$^{-2}$. The same behavior can also be seen in the region north-west of the T3 site. The aerosol cooling effect is mostly related with SOA production caused by the interaction between VOCs and NO$_x$. When the Manaus plume reaches regions downwind of the city, as seen on March 13, 2014, with few clouds, low precipitation and biomass burning, the plume has a cooling effect on the region as the plume evolves. The SWDNBC (clear sky) variable was used in this study to investigate the





aerosol radiative effect on the surface due limitations for simulate the cloud coverage on Amazonian region. Also, the results founded in this paper have to be investigated deeply in order to better understand the effects on the diffuse and direct radiation.

### 3.4    Aging Plume Impact on the Optical Properties

In this section, we examine how aging of the Manaus plume may affect its optical properties. SSA initially has low values of ca. 0.91, then increases after plume age 1 (7 LT). Some processes which affect SSA values as the plume ages are dilution, BC

deposition, SOA formation and the lensing effect (Holanda et al., 2020; Shrivastava et al., 2019; Cirino et al., 2018). The SSA values in the plume continue to increase during the plume aging process, consistent with SOA (ASOA + BSOA) production in the aging plume (see Fig. 11d). Our simulations show that, on March 13, 2014, the increase of SSA as the plume ages is mostly related to a combination of an increase in SOA formation and BC dilution. Figure 11f shows that BC and CO diluted in similar proportions, suggesting that, at this time scale, dilution is more important than deposition. When the plume is 3 hours old, total

organics reach ca. 11 $\mu g\ m^{-3}$ and at that time the plume is north of the T3 site (see Fig. S12d in the SI). Similar results were found by de Sá et al. (2018) at the T3 site.

During plume aging, a decrease of anthropogenic primary organic aerosol and a increase in SOA was observed, similar to results reported by Shilling et al. (2018). The biggest contribution to total SOA during the plume aging comes from anthropogenic emissions, ca. 70% of the total SOA. SOA production increases rapidly and saturates at plume age 4, indicating that

it is a challenge to represent these processes in tropical regions with global models, especially without correct treatments of sub-grid effects, such as the production of SOA. The simulated plume used in the tracking analysis traveled 160 km from Manaus (Fig. S13 in the SI). The distance between T1 and T3 is around 70 km, so when the plume reaches that distance from Manaus, it is ca. 3 hours old.

Figure 11b shows instantaneous downwelling clear sky shortwave flux at bottom with the Manaus plume turned on and off.

After 3 hours of plume aging, incoming solar radiation is reduced by ca. -15 W m$^{-2}$ and is further reduced by about -30 W m$^{-2}$ after the plume is 7 hours old. This reduction by 3% of solar flux and the resulting increase in diffuse radiation results in a significant increase in net primary productivity (Cirino et al., 2014; Rap et al., 2015). As the plume ages and dilutes, its impact on the solar radiation remains constant. Between hours 7 and 9, although the plume's attenuation of incoming solar radiation decreases in absolute terms, from ca. -32 W m$^{-2}$ to -27 W m$^{-2}$, as a percentage, the plume's attenuation remains constant at

ca. 3%.

BC simulations at an altitude of ca. 500 m above the ground were evaluated using aircraft measurements from the Manaus plume on March 13, 2014. For the most part, our simulation shows good agreement with the G1 measurements (SI Fig. S16, in the SI), particularly for background conditions. The offset in the third and fourth peaks is due to differences between the meteorological conditions of the simulation and reality. Similar offsets between simulations and observations were found by

Shrivastava et al. (2019).

As the plume ages SAE, begins to increase at 8 LT (after 2 hours of plume aging) and remains constant with values of ca. 1.17 until 13 LT (after 7 hours of plume aging). During this period, AAE is mostly close to 1, which can be explained by increased concentrations of fine (SOA > 15 $\mu g\ m^{-3}$) and absorbing (BC > 0.4 $\mu g\ m^{-3}$) particles near Manaus. Similar results



**Figure 11.** Simulated plume 100-500 m starting at Manaus 6 to 15 LT on March 13, 2014 in order to calculate the mean (a) SSA at 550nm, (b) Instantaneous downwelling clear sky shortwave flux at bottom (W m$^{-2}$), (c) Total organics ($\mu g \ m^{-3}$) normalized by $\Delta$CO, (d) Organic aerosol ($\mu g \ m^{-3}$), (e) SAE and AAE and (f) BC ($\mu g \ m^{-3}$) normalized by $\Delta$CO. The plume age represents the time the plume tracking started.





were found by Romano et al. (2019) in southeastern Italy from December 22, 2015 to March 30, 2016, with 1 < AAE < 1.5

and SAE > 1 using the classification defined by Cappa et al. (2016).

## 4   Summary and conclusions

Numerical simulations with the WRF-Chem model were performed in order to investigate the impact of the Manaus plume on aerosol optical properties downwind of Manaus and how the plume aging process can affect those optical properties. We use the simulations to investigate the impact of anthropogenic emissions on SOA formation over the Amazon region during the wet

season and the effect of anthropogenic $NO_x$ on $O_3$ production from $O_3$ precursors emitted by the forest. Aerosol characteristics have many impacts that could influence ecosystems on a regional scale. We selected March $13^{th}$, 2014 as a "golden day" (Shilling et al., 2018), to analyze the Manaus plume's influence at the T3 site and regions further downwind. During this day, the transport event brought elevated gas and aerosol concentrations from Manaus, associated with favorable meteorological conditions. According to our results, downwind of Manaus at the T3 site, the total organic aerosol mass increased by ca. 75%

(0.5 - 2.0 $\mu g\ m^{-3}$) when anthropogenic emissions were turned on. This increase in organic aerosol mass suggests the Manaus plume is primarily responsible for the changes in the physical and chemical aerosol population characteristics in those regions.

From model experiments, we conclude that the influence of the Manaus plume can reach areas up to 300 km downwind of Manaus, and provide a quantitative assessment of the effects urban pollution could cause to Amazonian forests surrounding urban centers. Overall, our simulations indicate that the aerosol impact of the Manaus plume increases irradiance values by

20% near the T3 site. We also separated the contributions of the different aerosol chemical components that contribute to our estimate of the total aerosol mass concentration and their impact on optical properties. Especially striking is the impact on $O_3$ formation. Due to the high $NO_x$ concentrations present in Manaus, the simulations showed that increased $O_3$ production mostly occurs in the regions to the south-west of Manaus, where an atmosphere conducive to $O_3$ enhancement can be found.

According to our results, the lowest $g_{aer}$ values were generally found in regions with a significant fraction of the aerosol load

coming from small size particles of anthropogenic origin, e.g., from TPPs and refineries in the Manaus region. Conversely, the largest $g_{aer}$ values were observed over regions with aerosol dominated by large particles of biogenic origin (T0a site). Further investigations are necessary to determine if different sulfate amounts from anthropogenic emissions may change the strong direct effect for high aerosol particle concentrations. More ground-based aerosol and trace gas observations over the western Amazon region could help to evaluate the magnitude of the aerosol effect in this area.

This study contributes to the investigation of the optical properties of $PM_{2.5}$ over the Amazon region during the wet season. To assess the impact of Manaus emissions on SOA production, and consequently, on aerosol optical properties, WRF-Chem model runs were conducted with and without anthropogenic emissions. Assuming only biogenic emissions with boundary conditions from the global model, OA production decreased by 75% at the T3 site. This study also shows that on March 13, 2014, the aerosol aging process caused a gradual increase in SSA. Additionally, due to the deposition process, significantly

decreasing concentrations of BC are found during plume evolution. This process, combined with SOA formation, contributes to the increase in SSA as the plume ages. The results of this study demonstrate that uncertainties in coating processes of organic





aerosols involving BC particles also warrant additional study to better account for a possible decrease in SSA during the plume aging process. The results here also demonstrate that in order to precisely calculate the radiative forcing impact, it is important to take into account all SOA formation mechanisms, including VOC oxidation, especially for tropical forest regions like the Amazon. One action that may improve SOA model accuracy is to update some of the MEGAN model inputs when new data such as emission factors and vegetation coverage data becomes available. In addition, there are very few long term aircraft based ASOA and BSOA observations data and more observations could help validate the models and improve their accuracy.

*Data availability.* The GoAmazon2014/5 experiment data are available from the ARM website: https://www.arm.gov/research/campaigns/ amf2014goamazon and from the Laboratory of Atmospheric Physics - LFA website: http://lfa.if.usp.br/ftp/public/LFA_Processed_Data/. The simulations and analysis code generated for this study are available upon request from JPN.

*Author contributions.* JPN, MMB and PA conceptualized and defined the methodology. JPN carried out the formal analysis and investigation of the model results with support from BM, MMB, ALB, HG, LVR, SC, HJB, MAF, MT, SAM and PA. ALVV, SAAR, HG and MMB supported the design and running of simulations. GGC, PA, LVR, MLB, BM and RAFS collected and curated the experimental data. JPN wrote the original draft and all authors discussed the results and commented on the paper.

*Competing interests.* The authors declare that they have no conflict of interest.

*Acknowledgements.* We acknowledge support from the Central Office of the Large-Scale Biosphere Atmosphere Experiment in Amazonia (LBA), coordinated by the National Institute of Amazonian Research (INPA) and the Amazonas State University (UEA), Amazonas, Brazil. JPN thanks the Brazilian Federal Agency for Support and Evaluation of Graduate Education (CAPES) for a graduate fellowship, linked to the doctoral program in Climate and Environment (CLIAMB) and for supporting 7 months of a visiting graduate student program at the NOAA Earth System Research Laboratory. JPN also thanks the Institute of Physics of the University of São Paulo (IFUSP) for student mobility and logistical support, and CIRES and NOAA ESRL for financial and logistical support. We thank Michael Trainer for providing support and knowledge during the research. We thank Manish Shrivastava for providing WRF-Chem simulation output for comparison with this work. We thank Steven Jefferts, Stefania Romisch and Samuel Brewer for facilitating communication between members of this collaboration. We are grateful to Bruno Takeshi, Luiz Cândido, Renata Teixeira and Delano Campos for instrument operation and data analysis. Finally, we thank Richard Tisinai for IT support. MAF acknowledges a scholarship from CNPq, Project 169842/2017-7, for supporting his PhD studies at the IFUSP, São Paulo, Brazil, and CAPES, Project 88887.368025/2019-00, for supporting 6 months of a visiting graduate student program at the Max Planck Institute for Chemisty, Mainz, Germany. BM acknowledges a scholarship from CNPq, Project 133393/2019-4, for supporting his Masters studies at the IFUSP, São Paulo, Brazil. HG acknowledges funding from CAPES through grant 1757/2017. PA acknowledges funding from FAPESP through grant 2017/17047-0.





*Financial support.* JPN has been supported by the Brazilian Federal Agency for Support and Evaluation of Graduate Education (CAPES) - Project Code 88882.444345/2018-01 and 88881.190103/2018-01.



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
