# Peer review of "Aerosols from anthropogenic and biogenic sources and their interactions: modeling aerosol formation, optical properties and impacts over the central Amazon Basin"

_Atmospheric Chemistry and Physics, 2020_

## Referee Comment (RC1) · Anonymous Referee #1 · 4 Dec 2020

Journal

Atmospheric Chemistry and Physics

Title

Aerosols from anthropogenic and biogenic sources and their interactions: modeling aerosol formation, optical properties and impacts over the central Amazon Basin

[Figure]

Authors

Janaína P. Nascimento, Megan M. Bela, Bruno Meller, Alessandro L. Banducci, Luciana V. Rizzo, Angel Liduvino Vara-Vela, Henrique M. J. Barbosa, Helber Gomes, Sameh A. A. Rafee, Marco A. Franco, Samara Carbone, Glauber G. Cirino, Rodrigo A. F. Souza, Stuart A. McKeen, and Paulo Artaxo

Summary

This study analyses how biogenic and anthropogenic air pollution impact aerosol formation and optical properties over the Amazon. The paper uses campaign measurements from the Green Ocean Amazon experiment (GoAmazon2014/5) and high-resolution chemical transport model simulations with and without anthropogenic emissions from Manaus. The paper is well written. The paper is suitable to the scope of Atmospheric Chemistry and Physics. My main criticism is that it would be useful to add a more detailed discussion of the implications of these findings.
Overall, this paper is a high quality and detailed investigation of aerosols over the Amazon with interesting findings. The reach of the paper would be improved from further contextualising the results.

Comments

1. Lines 14, 138, and Table 2: Define acronym at first use.

2. Figure 1 is high quality.

3. Line 140: The lowest volatility bin has a saturation concentration of 1 $\mu$g m$^{-3}$. Does this not exclude lower volatility compounds (e.g. 0.1 $\mu$g m$^{-3}$), which could be potentially important e.g. Shilling et al., (2008)?

4. Lines 191-193: Supplementary Figure 2 shows that the model underestimates precipitation. The implications of this is not discussed in the meteorlogical analyses.

5. The small font size within some plots is difficult to read e.g. Supplementary Figure 4, Figure 10.

6. Line 311: Typo: Additional bracket.

7. Supplementary Figure 9 is not referenced in the main text.

8. The ordering of some Supplementary Figures (e.g. 10, 11, 12, 14, 15) does not match how they are referenced in the main text.

9. Figure 8: The rainbow colour bar may be difficult to distinguish for some readers, and I suggest using an alternative e.g. ColorBrewer 2.0.

10. Figure 8: Typo: subplot (c) named subplot (b).

11. Line 502: Observations evaluate models, not validate.

12. If the Figure captions are to be read or searched independently of the main paper, then define the acronyms they contain.

References

Shilling, J. E., Chen, Q., King, S. M., Rosenoern, T., Kroll, J. H., Worsnop, D. R., McKinney, K. A. and Martin, S. T.: Particle mass yield in secondary organic aerosol formed by the dark ozonolysis of $\alpha$-pinene, Atmos. Chem. Phys., 8(7), 2073–2088, doi:10.5194/acp-8-2073-2008, 2008.

---

## Referee Comment (RC2) · Anonymous Referee #2 · 15 Jan 2021

The paper presents a case study of observational and modeled data of the impact of the urban plume of the city of Manaus in the Amazon in Brazil on the pristine regions downwind of the city within the rainforest. The WRF-CHEM model simulates atmospheric natural and anthropogenic aerosols and trace gases and investigate the formation of Secondary Organic Aerosols as well as ozone. In general, the manuscript is very well written, is based on the relevant literaturem presents interesting new results and fits within the scope of ACP.

In some parts the discussion lacks a bit of clarity and I suggest that the findings be

communicated in a clearer way, prior to publication.

In the following I list general comments that should be addressed. The specific comments can be found in the attached commented pdf-file.

1.) In the meteorology section, the general meteorological patterns of the region could be described in a bit more detail, for readers who are not familiar with the region.

2.) During the text you vary between present and past tense , revise for consistency.

3.) As you solely analyze one episode, how representative is it in general for the region? Is the plume of Manuas always heading in this direction?

4.) It is not explained why you needed to use the HYSPLIT model, when you have a powerful 3D atmospheric-chemistry model at hands, this should be better explained.

5.) You report an offset of three hours between peaks of your modeled data versus observed data for the meteorological parameters, which is not so small. If this has not yet been done check in detail if both data sets are either in UTC or LT. The statistics in numbers looks good, but the hight Pearson coefficient with this offset seems unlikely, please confere.

6.) Why are there so significant differences between the BC of the global ECMWF model and your WRF runs, if the regional WRF model uses the global model data as initial and boundary conditions (Figure 2)? The global model should provide the adequate level of BC concentrations from the trans-atlantic transport into the domain of the regional model via its north/eastern domain border.

Please also note the supplement to this comment:
https://acp.copernicus.org/preprints/acp-2020-1002/acp-2020-1002-RC2-
supplement.pdf

**Supplement:**

[revised manuscript text omitted]

---

## Author Comment (AC1) · 7 Feb 2021

Initially, we want to thank the reviewers for their insightful and constructive comments that have significantly improved the revised version of the manuscript. Reviewer 1 comment's on improving the conclusions and providing a broader aspect of our results was very welcome. Comments from reviewer 2 helped to clarify some methodological issues that also improved the revised version.

In order to facilitate the identification of actions for each individual review's comments,

we coded reviewers identification as R1 and R2 (Reviewer #1 and #2). Answers for each individual comment (C) are numbered for each reviewer: R1-C1, R2-C2, etc.

A copy of these author comments with their original formatting is attached as the supplement to this comment.

Reviewer #1: This study analyses how biogenic and anthropogenic air pollution impact aerosol formation and optical properties over the Amazon. The paper uses campaign measurements from the Green Ocean Amazon experiment (GoAmazon2014/5) and high-resolution chemical transport model simulations with and without anthropogenic emissions from Manaus. The paper is well written. The paper is suitable to the scope of Atmospheric Chemistry and Physics. My main criticism is that it would be useful to add a more detailed discussion of the implications of these findings. Overall, this paper is a high quality and detailed investigation of aerosols over the Amazon with interesting findings. The reach of the paper would be improved from further contextualising the results.

Thanks for the good suggestion, we now realize the need to clarify our findings and contributions to the efforts in this particular research area. We have rewritten or modified parts of the conclusions section in order to include a more detailed discussion of the implications of our findings. The changes can be found in the main text (Line 495 – 559, revised version).

R1-C1: Lines 14, 138, and Table 2: Define acronym at first use We have defined the acronyms on first use. The changes can be found in the main text (Lines 15 and 162, revised version). We also changed Table 2 (Page 12, revised version) and various other acronyms.

R1-C2: Figure 1 is high quality.

Thank you very much.

R1–C3: Line 140: The lowest volatility bin has a saturation concentration of 1 ug/m3.

Does this not exclude lower volatility compounds (e.g. 0.1 $\mu$g/m3), which could be potentially important e.g. Shilling et al., (2008)?

That is a great question. The VBS approach within the WRF/Chem option used in this work, uses the Murphy and Pandis (2009) SOA yields for the various biogenic and anthropogenic VOC into 4 distinct saturation bins (1, 10, 100, 1000 $\mu$g/m3). The laboratory data, and parametric fitting used in that study is specific to those 4 size bins. So, there is no explicit accommodation for SOA species with equivalent saturation concentrations of less than 1 $\mu$g/m3. The text has been modified to make this point clearer (Line 167, revised version), and a reference to this possible limitation with the Schilling et al., (2008) paper you suggest has been added. It would be better to have bins at 0.1 $\mu$g/m3 in order to better fit the VBS model, but with the VBS model we have now, we are able to extrapolate volatilities below our bins. See Fig 3 of Kroll and Seinfeld et al 2008. Furthermore, there is experimental difficulty in determining mass loadings for volatilities below 1 ug/m3 (Shrivastava et al, 2019). If the mass loadings for OA species beyond $\alpha$-pinene for 0.1 $\mu$g/m3 could be determined it would improve the VBS method, but currently, the best determination of the VBS parameters for a wide range of OA are given in Ahmadov et al, 2012 with the (1,10,100,1000) bined experimental data.

R1–C4: Lines 191-193: Supplementary Figure 2 shows that the model underestimates precipitation. The implications of this are not discussed in the meteorological analyses.

The model is indeed underestimating the total amount of the precipitation during the simulated days (Fig S2). However, these 4 days were chosen because they show very little precipitation, compared to the average wet season in Central Amazonia. Because of this, precipitation had quite a small impact on the chemistry on March 12 and 13th at the T3 site, especially in the morning. Hence, we don't expect this precipitation bias to affect our atmospheric chemistry simulations very much. The text has been modified to make this point clearer (Line 226 - 231, revised version). Having said that, we point out that modeling the Amazonian precipitation during the wet season is not an easy

task. WRF has some options for microphysics and the choice depends on sensibility tests in order to find the best options for each case. A recent paper for the Central Amazon (Sátyro et al. 2020) performs such microphysics sensibility study, and their best configuration shows an underestimation of precipitation similar to ours.

R1–C5: The small font size within some plots is difficult to read e.g. Supplementary Figure 4, Figure 10. The font size for Fig 4 and Fig 10 were changed in order to be more legible.

R1–C6: Line 311: Typo: Additional bracket.

The additional bracket was removed.

R1–C7: Supplementary Figure 9 is not referenced in the main text.

Thanks for your comment, we are sorry for our mistake. The Figure is now 12 and referenced in the main text (Line 400, revised version).

R1–C8: The ordering of some Supplementary Figures (e.g. 10, 11, 12, 14, 15) does not match how they are referenced in the main text. Again, we apologize for our mistake. The ordering of the figures now matches their references in the main text.

R1–C9: The rainbow colour bar may be difficult to distinguish for some readers, and I suggest using an alternative e.g. ColorBrewer 2.0.

Thank you for suggesting the use of ColorBrewer 2.0 to improve legibility of our plots. Figure 8 was modified to allow an easier distinction of the different components. We used the colormap Viridis.

R1–C10: Figure 8: Typo: subplot (c) named subplot (b).

Thanks for your observation. The subplot name in Figure 8 was changed.

R1–C11: Line 502: Observations evaluate models, not validate

Yes, you are right about the term. We have changed it in the main text (Line 546,

revised version).

R1–C12: If the Figure captions are to be read or searched independently of the main paper, then define the acronyms they contain.

Agreed. We have modified the figures (2, 4, 5, 6, 7, 8 and 9) captions in the revised version.

References:

Ahmadov, R., McKeen, S., Robinson, A., 530 Bahreini, R., Middlebrook, A., De Gouw, J., Meagher, J., Hsie, E.-Y., Edgerton, Shaw, S., and Trainer, M.: A volatility basis set model for summertime secondary organic aerosols over the eastern United States in 2006, J. Geophys. Res., 117, D06301, https://doi.org/10.1029/2011JD016831, 2012.

Murphy, B. N., and S. N. Pandis (2009), Simulating the formation of semi volatile primary and secondary organic aerosol in a regional chemical trans- port model, Environ. Sci. Technol., 43(13), 4722–4728, doi:10.1021/ es803168a.

Kroll, J. H. and Seinfeld, J. H.: Chemistry of secondary organic aerosol: Formation and evolution of low-volatility organics in the atmosphere, Atmos. Environ., 42, 3593–3624, 2008.

Sátyro, Zayra Christine, et al. "The relative and joint effect of rivers and urban area on a squall line in the Central Amazonia." Science of the Total Environment 755 (2020): 142178.

Shrivastava, M., Andreae, M. O., Artaxo, P., Barbosa, H. M., Berg, L. K., Brito, J., Ching, J., Easter, R. C., Fan, J., Fast, J. D., et al.: Urban pollution greatly enhances formation of natural aerosols over the Amazon rainforest, Nature communications, 10, 1–12, https://doi.org/10.5194/acp-17-7977-2017, 2019.

Reviewer #2: The paper presents a case study of observational and modeled data of the impact of the urban plume of the city of Manaus in the Amazon in Brazil on the

pristine regions downwind of the city within the rainforest. The WRF-CHEM model simulates atmospheric natural and anthropogenic aerosols and trace gases and investigate the formation of Secondary Organic Aerosols as well as ozone. In general, the manuscript is very well written, is based on the relevant literature presents interesting new results and fits within the scope of ACP. In some parts the discussion lacks a bit of clarity and I suggest that the findings be communicated in a clearer way, prior to publication.

Thanks for your good suggestions. We have rewritten and modified the main text in order include your suggestions, especially to clarify our main findings in a clearer way.

R2–C1: In the meteorology section, the general meteorological patterns of the region could be described in a bit more detail, for readers who are not familiar with the region.

Thanks for your suggestion. We have modified the main text (Line 83 - 94, revised version) in order to describe the general meteorological patterns of the region with more detail.

R2 – C2: During the text you vary between present and past tense, revise for consistency.

We apologize for these language issues. We have modified the text in order to keep the consistency.

R2 - C3: As you solely analyze one episode, how representative is it in general for the region? Is the plume of Manaus always heading in this direction?

Thanks for your really good point. The climatological large-scale winds blow consistently from the east in the wet season. This was a critical issue in the planning stages of the GoAmazon14/5 experiment, to allow sampling in- and out-plume at T3, and is discussed in detail in Martin et al., 2016, and Martin et al., 2017). During the GoAmazon wet season campaign, meteorological characterization was carefully measured using several ground-based cloud and meteorological radars and large-scale simulations. Those results showed that the initial assessment was correct, i.e., that the wind blows from Manaus towards T3 (Cirino et al., 2018; Martin et al., 2016). The simulation period (8-15 Mar 2014) was chosen because of relatively low precipitation, with a low amount of squall lines reaching the site, and no contribution from biomass burning. We emphasize that this "golden week" was also chosen by other GoAmazon studies such as (Rafee et al. 2017; Shrivastava et al. 2019) for the same reasons.

The day March 13th, in particular, is a "golden day" because it has steady winds during the daytime, few clouds, mostly sunny skies and no precipitation (Shilling et al. 2018). This avoids the complex meteorology that would be expected from River-breeze circulation or convective system, which is discussed in detail in papers dealing with the chemistry-meteorology connection such as (Cirino et al. 2018; de Sá et al., 2018; Palm et al., 2017). For these reasons, the day we focused on can be regarded as a "characteristic" wet season sunny day. The discussion on this issue was expanded on line 107 - 112 on the revised version.

R2 - C4: It is not explained why you needed to use the HYSPLIT model, when you have a powerful 3D atmospheric-chemistry model at hands. This should be better explained.

Thanks for your comment. WRF-Chem indeed produces the 3D atmospheric state, including meteorology and chemistry. However, WRF-Chem does not provide a time-axis for each air-parcel, and all pollutants (e.g. CO) packaged together depending on their source (e.g CO emitted from Manaus is added to the background CO). Therefore, it is not easy to investigate how does the Manaus plumes ages. The use of HYSPLIT forced by the WRF-Chem simulated winds help us to visualize the plume trajectory, facilitates the chemical tracing methodology, and allows calculating the aging of the plume as we get a time stamp on the trajectory points.

Doing this with WRF-Chem only would require modifying the source-code to add a passive tracer that would be emitted only on March 13th from 6am to 7am. Then, by following this tracer we would locate the movement of the encompassing air mass, and

hence derive the spreading of the plume and a time-axis for its movement. Alternately, a chemical tagging scheme could be developed, allowing us to isolate every compound by its source and emission time. Both of these strategies are much more complicated than running HYSPLIT on the WRF-Chem output and would not be feasible without the help of the WRF-Chem developers. The discussion on this issue was expanded on line 116 – 121 on the revised version

R2 - C5: You report an offset of three hours between peaks of your modeled data versus observed data for the meteorological parameters, which is not so small. If this has not yet been done check in detail if both data sets are either in UTC or LT. The statistics in numbers looks good, but the High Pearson coefficient with this offset seems unlikely, please confere.

We thank your suggestion. We double checked the time zones and the results presented in the manuscript are correct.

R2 - C6: Why are there so significant differences between the BC of the global ECMWF model and your WRF runs, if the regional WRF model uses the global model data as initial and boundary conditions (Figure 2)? The global model should provide the adequate level of BC concentrations from the transatlantic transport into the domain of the regional model via its north/eastern domain border.

This is a really good point. We believe the differences are related with the changes in the mixing and deposition mechanisms and in model resolution. In addition, differences in emission schemes between the WRF-Chem and ECMWF models can influence, through transport, the BC concentrations not only at the T0a site but in general. According to previous work (Moran-Zuloaga et al.,2018) the T0a site receives sporadic air masses loaded with marine aerosol transported from the Atlantic Ocean, and dust outflows from the Sahara Desert, together with smoke from fires in West Africa (Ben-Ami et al., 2010; Andreae et al., 2012, 2015; Rizzolo et al., 2017; Pöhlker et al., 2018). Within the data coming from the global ECMWF model (Figure 2), it is possible to observe that, on the 10th and 11th of March 2014, BC (both simulated and observed) were above expected levels (0.1 $\mu$g/m3), consistent with coherent long-range transport of BC from west Africa (Moran-Zuloaga et al., 2018). Those two day were representative of BC transport from Africa where the episodes depend on the Inter Tropical Convergence Zone (ITCZ) positioning, as well as the air mass trajectories from Africa to the Central Amazon. This explains the relatively high BC concentrations reported.

References

Rafee, S. A. A., Martins, L. D., Kawashima, A. B., Almeida, D. S., Morais, M. V., Souza, R. V., Oliveira, M. B., Souza, R. A., Medeiros, A. S., Urbina, V., et 750 al.: Contributions of mobile, stationary and biogenic sources to air pollution in the Amazon rainforest: a numerical study with the WRF-Chem model, Atmos. Chem. Phys., 17, 7977, https://doi.org/10.5194/acp-17-7977-2017, 2017.

Shrivastava, M., Andreae, M. O., Artaxo, P., Barbosa, H. M., Berg, L. K., Brito, J., Ching, J., Easter, R. C., Fan, J., Fast, J. D., et al.: Urban pollution greatly enhances formation of natural aerosols over the Amazon rainforest, Nature communications, 10, 1–12, https://doi.org/10.5194/acp-17-7977-2017, 2019.

Shilling, J. E., Pekour, M. S., Fortner, E. C., Artaxo, P., Sá, S. d., Hubbe, J. M., Longo, K. M., Machado, L. A., Martin, S. T., Springston, S. R., et al.: Aircraft observations of the chemical composition and aging of aerosol in the Manaus urban plume during GoAmazon 2014/5,785 Atmos. Chem. Phys., 18, 10 773–10 797, https://doi.org/10.5194/acp-18-10773-2018, 2018.

de Sá, S. S., Palm, B. B., Campuzano-Jost, P., Day, D. A., Hu, W., Isaacman-VanWertz, G., Yee, L. D., Brito, J., Carbone, S., Ribeiro, I. O., Cirino, G. G., Liu, Y. J., Thalman, R., Sedlacek, A., Funk, A., Schumacher, C., Shilling, J. E., Schneider, J., Artaxo, P., Goldstein, A. H., Souza, R. A. F., Wang, J., McKinney, K. A., Barbosa, H., Alexander, M. L., Jimenez, J. L., and Martin, S. T.: Urban influence on the concentration and composition of submicron particulate matter in central Amazonia, Atmos. Chem. Phys.,

18, 12 185–12 206, https://doi.org/10.5194/acp-18-12185-2018, 2018.

Palm, B. B., de Sá, S. S., Day, D. A., Campuzano-Jost, P., Hu, W., Seco, R., Sjostedt, S. J., Park, J.-H., Guenther, A. B., Kim, S., et al.: Secondary organic aerosol formation from ambient air in an oxidation flow reactor in central Amazonia, Atmos. Chem. Phys. Discussions (Online), 18, https://doi.org/10.5194/acp-18-467-2018, 2018.

Martin, S., Artaxo, P., Machado, L., Manzi, A., Souza, R., Schumacher, C., Wang, J., Andreae, M., Barbosa, H., Fan, J., et al.: Introduction: observations and modeling of the Green Ocean Amazon (GoAmazon2014/5), Atmos. Chem. Phys., 16, https://doi.org/10.5194/acp-16- 4785-2016, 2016

Martin, S. T., Artaxo, P., Machado, L., Manzi, A. O., Souza, R., Schumacher, C., Wang, J., Biscaro, T., Brito, J., Calheiros, A., et al.: The Green Ocean Amazon experiment (GoAmazon2014/5) observes pollution affecting gases, aerosols, clouds, and rainfall over the rain forest, Bulletin of the American Meteorological Society, 98, 981–997, https://doi.org/10.1175/BAMS-D-15- 00221.1, 2017.

Response to the Reviewer 2 comments in the annotated PDF file

We thank really much the reviewer for the additional comments in the annotated PDF which helped us to improve our paper. All the observations in the supplement file were taken into account.

1- Also describe the semi-direct effect below.

The semi-direct effect is now described on the main paper (Line 33 - 35, revised version).

2- Are you really exclusively focusing on optical properties, or also on physical properties?

The objective of this work is to model secondary aerosol formation in Central Amazonia, comparing modeled scenarios with and without anthropogenic emissions, examining the interactions between natural biogenic emissions and urban air pollution from Manaus and investigating their impact on aerosol optical properties. Our simulations consider both physical and optical aerosol properties, but the primary focus is on the interaction between the chemistry of the aerosol with the radiation and the process of the aging plume.

3- what do you mean? (in ca. 8 km distance downwind from T1"?

We mean the distance of the T1 site from downtown Manaus (latitude 3_06'07" and longitude 60_01'30"). We rewrote the sentence to make it clearer.

4- Is this enough to achieve chemical equilibrium?

Previous studies have considered the first one (Rafee et al. 2017; Medeiros et al. 2017) or two (Vara-Vela et al., 2018) days as spin-up time. In our results, we focused on days between 10th to 13th, excluding the first two days from our simulations.

5- "golden day"

This term golden day is used due to sunny conditions and clear evolution of the plume downwind of Manaus on March 13. Due the comment R2 - C3 it was better explained on lines 107 - 112 on the revised version.

6- Why are you looking at CO if your focus are aerosols?

We use CO as a passive tracer. It is a common choice as a tracer because has a long residence time in the atmosphere (much longer than the transport time of the Manaus plume) and it is almost entirely anthropogenic in origin, emitted during combustion and other anthropogenic processes. In addition, it is significantly enhanced in urban plumes relative to the background, and it is routinely and robustly measured (Shilling et al. 2018; Shrivastava et al. 2019).

7- which data set exactly European Centre for Medium-Range Weather Forecasts (ECMWF)?

https://atmosphere.copernicus.eu/configuration-daily-global-analyses-and-forecasts
https://apps.ecmwf.int/datasets/data/cams-nrealtime/levtype=sfc/

8- These measurements were not performed in the framework of this study, right? On this case, this should be clearly stated. For example, you can say that the "available databases are this and that..."

Yes, the measurements were not taken in the scope of this work, as it was described on section 2.3.1 (revised version) The availability of the data is described in section data availability (line 562 – 564. Revised version).

9- Why do you regard 10 m wind speed, and not at 2 m?

We did so because the typical altitude for measuring surface wind is 10 m (https://library.wmo.int/doc_num.php?explnum_id=7372), and GoAmazon weather station measured wind at that height.

10- The Saharan dust is mineral dust, not BC. However, you can have mixed aerosols of mineral dust, with a BC coating, when dust plumes mix with the fire plumes of the N-HEM African fires that also occur from December-February, mainly (when the ITCZ is further to the south).

That is a good question. According to Pöhlker et al., 2019, the backward trajectory from 1 to 31 March 2014, characteristic of the Amazonian wet season conditions, shows that the plumes transported in the rainy season are not exclusively from the Sahara dust, there are also the contributions from the fires coming in West Africa. Figure 1a presented in Pöhlker et al., 2019 shows the HYSPLIT backward trajectory that reach ATTO between 01 to 31 March 2014.

11- Is it also possible that you have fire aerosols reaching the Amazon from S-Hem Africa from May-July?

Thanks for your interesting question. According to Figure 12 in Holanda et al., 2020, the period between May – July is a transition state and from July to October it is possible

to see very relevant contributions coming from biomass burning. Section 2.6 (Holanda et al., 2020) shows how to calculate these fire events.

References:

Medeiros, A. S. S. et al. Power plant fuel switching and air quality in a tropical, forested environment. Atmos. Chem. Phys. 17, 8987–8998 (2017).

Holanda, B. A., Pöhlker, M. L., Saturno, J., Sörgel, M., Ditas, J., Ditas, F., Wang, Q., Donth, T., Artaxo, P., Barbosa, H. M. J., Borrmann, S., Braga, R., Brito, J., Cheng, Y., Dollner, M., Kaiser, J. W., Klimach, T., Knote, C., KruÌĹger, O. O., FuÌĹtterer, D., LavricËĞ, V. J., Ma, N., Machado, L. A. T., Ming, J., Morais, F. G., Paulsen, H., D., S., Schlager, H., Schneider, J., Su, H., Weinzierl, B., Walser, A., Wendisch, M., Ziereis, H., Zöger, M., Pöschl, U., Andreae, M. O., and Pöhlker, C.: Influx of African biomass burning aerosol during the Amazonian dry season through layered transatlantic transport of black carbon-rich smoke, Atmos. Chem. Phys., 20, 4757–4785, 2020.

Pöhlker, C., Walter, D., Paulsen, H., Könemann, T., Rodriguez-Caballero, E., Moran-Zuloaga, D., Brito, J., Carbone, S., Degrendele, C., Després, V. R., Ditas, F., Holanda, B. A., Kaiser, J. W., Lammel, G., LavriËĞc, J. V., Ming, J., Pickersgill, D., Pöhlker, M. L., Saturno, J., Sörgel, M.,Wang, Q.,Weber, B.,Wolff, S., Artaxo, P., Pöschl, U., and Andreae, M. O.: Land cover and its transformation in the backward trajectory footprint region 775 of the Amazon Tall Tower Observatory, Atmos. Chem. Phys., 19, 8425–8470, https://doi.org/10.5194/acp-19-8425-2019, 2019.

Please also note the supplement to this comment: https://acp.copernicus.org/preprints/acp-2020-1002/acp-2020-1002-AC1-supplement.pdf